## Evidence synthesis

health and disease and epidemiology

England primary school COVID-19 risks, schools opening, stochastic uncertainty analysis, Bayesian belief network, scenario sensitivity tests

**Author for correspondence:**
W. P. Aspinall
e-mail: willy.aspinall@bristol.ac.uk

# Pupils returning to primary schools in England during 2020: rapid estimations of punctual COVID-19 infection rates

W. P. Aspinall[1,2], R. S. J. Sparks[1], M. J. Woodhouse[1,3], R. M. Cooke[4,5], J. H. Scarrow[6] and the CoMMinS Project[7]

[1]School of Earth Sciences, University of Bristol, Bristol BS8 1RJ, UK
[2]Aspinall and Associates, Tisbury SP3 6HF, UK
[3]School of Mathematics, University of Bristol, Bristol BS8 1QU, UK
[4]Institute of Applied Mathematics, Delft University of Technology, Building 28, Mourik Broekmanweg 6, 2628 XE Delft, The Netherlands
[5]Resources for the Future, 1616 P Street NE, Washington, DC, USA
[6]Departamento de Mineralogía y Petrología, Facultad de Ciencias, Universidad de Granada, 18071 Granada, Spain
[7]CoMMinS Project 'COVID-19 Mapping and Mitigation in Schools', Bristol Medical School (PHS), University of Bristol, BS8 2PS Bristol, UK

WPA, 0000-0001-6014-6042; RSJS, 0000-0001-7173-2899; MJW, 0000-0002-2198-6791; RMC, 0000-0003-0643-1971; JHS, 0000-0001-8585-8679

Drawing on risk methods from volcano crises, we developed a rapid COVID-19 infection model for the partial return of pupils to primary schools in England in June and July 2020, and a full return in September 2020. The model handles uncertainties in key parameters, using a stochastic re-sampling technique, allowing us to evaluate infection levels as a function of COVID-19 prevalence and projected pupil and staff headcounts. Assuming average national adult prevalence, for the first scenario (as at 1 June 2020) we found that between 178 and 924 [90% CI] schools would have at least one infected individual, out of 16 769 primary schools in total. For the second return (July), our estimate ranged between 336 (2%) and 1873 (11%) infected schools. For a full return in September 2020, our projected range was 661 (4%) to 3310 (20%) infected schools, assuming the same prevalence as for 5 June. If national prevalence fell to one-quarter of that, the projected September range would decrease to between 381 (2%) and 900 (5%) schools but would increase to between 2131 (13%) and 9743 (58%) schools if prevalence increased to 4× June level. When regional variations in

prevalence and school size distribution were included in the model, a slight decrease in the projected number of infected schools was indicated, but uncertainty on estimates increased markedly. The latter model variant indicated that 82% of infected schools would be in areas where prevalence exceeded the national average and the probability of multiple infected persons in a school would be higher in such areas. *Post hoc*, our model projections for 1 September 2020 were seen to have been realistic and reasonable (in terms of related uncertainties) when data on schools' infections were released by official agencies following the start of the 2020/2021 academic year.

## 1. Introduction

On 11 May 2020, the UK government announced that selected primary school-age children in England would return to school on 1 June 2020. The returning groups would include Reception, Year 1 and Year 6 children, noting that there would also be some Year 2 to Year 5 children of frontline workers and those identified as vulnerable who were already being taught in many schools. With some nurseries being part of primary schools, Nursery age children were also expected to return. In the process, the UK government abandoned a proposal for a full return of all primary school children in England before the summer holidays; the expectation was that all schools (Primary and Secondary) would fully re-open for the new academic year, in early September 2020 (we adopt 1 September as a nominal start date).

The partial re-opening of primary schools was widely debated with concerns expressed by some parents, by teaching unions and teacher associations about the safety of the children and school staff. There were also concerns about the effect of opening schools on infection incidence in the wider community, for example by triggering a second wave. While official guidelines for re-opening schools were issued by the Department of Education (DfE) and formed the policy basis for risk mitigation in schools, some individual schools adopted their own bespoke strategies [1].

Here, we distinguish between societal risk, operational risk and individual risk. The question of how opening schools affects the national and regional picture of infection would be an example of societal risk assessment, for instance, by estimating the change in the reproduction number (e.g. [2]). Operational risk concerns a school identifying and managing an infection or the threat of an infection breakout. Mitigating steps might include asking confirmed or suspected COVID-19 persons to self-isolate, classes being sent home or even the whole school closing again. Individual risk concerns the probability that a child, teacher or other support staff acquires infection from another, perhaps asymptomatic, infected person in the school.

This study formed part of a quantitative hazard assessment of the threat of encountering COVID-19 infections in primary school pupils and staff. In support of this, we transferred numerical hazard and risk methods—developed for protecting people threatened by volcanic eruptions [3]—to enumerate potential infection levels in primary schools, taking formal account of estimates of uncertainties associated with attendance and prevalence factors. Our approach involved building a stochastic uncertainty model of hazard which includes information on schools and epidemiological parameters that control the occurrence of COVID-19 within schools. To the extent possible, and recognizing several limitations existed, information was derived from available contemporary national statistics on schools and on the incidence and prevalence of COVID-19 in England (i.e. as of May–June 2020).

We developed a quick, punctual model for enumerating infection hazard levels in schools, here defined as the probability of one (or more) infectious person (child, teacher or support staff) being on the premises of a school on any one day. The study used an efficient stochastic uncertainty modelling tool to construct an estimator for infection hazard levels in primary schools in England, with the intention this would be a first step in developing a risk model including infection transmission.

Some schools re-opened on 1 June 2020, some delayed their re-start while, in other cases, schools did not re-open under advice from local education authorities. Data from DfE indicates that on 2 July 2020 about 88% of state-funded primary schools had re-opened to some extent. The public response was variable and between 1 and 15 June 2020, approximately 35% of eligible children returned. In the following month, the numbers increased moderately: 41% of Year 1 pupils and 49% (Year 6) attended on 2 July. Between 18 May and 31 July 2020, there were 247 COVID-19 related incidents in schools, of which 116 resulted in positive tests [4].

Thus, we were able to follow up on our initial projections concerning the numbers of pupils' returning to primary schools in England in June–July 2020 and the potential numbers of infection incidents, and compare these with surveillance data reported by various government agencies. Similar comparisons were possible for our projections presaging the planned full return to schools in September 2020, for

the start of the 2020/2021 academic year. Given the extensive uncertainties involved, in both instances our forecast assessments were persuasively corroborated; we report these comparisons.

Our COVID-19 infection hazard model was extended to take account of regional variations in prevalence to characterize infection hazard level in relation to geographical locations. However, we did not address the implications of opening schools for the spread of the infection within the local or wider community; that was the objective of other modelling studies.

The model is flexible and amenable to rapid updating as new data arrives. We envisaged it would allow risk to be assessed for different return-to-school scenarios, including secondary schools, for regional or local situations and potentially for individual schools. Our approach can also be configured for different kinds of educational operation, such as universities and further education colleges, and it is currently being developed to inform the sampling strategy and programme for a project of infection testing in selected schools [5].

# 2. Hazard and risk terminology

Different domains of science and society use terms like 'hazard' and 'risk' in different ways and in different contexts. Risk can be discussed as, say, lives endangered by a natural disaster, whereas in the financial sector risk is defined as expected monetary loss. However, in the context of public health and safety, the risk is the chance of harm whereas, in some other contexts, 'threat' is used to signal the potential for harm. 'Exposure' is sometimes used to indicate that an asset or person is in harm's way. Risk also pertains to different scales and kinds of loss or harm. Therefore, risk management benefits from information on the hazard, on exposure, on vulnerability to the hazard and on its potential impact. Ultimately, quantitative risk assessment involves calculating the probability of a loss or a harm. Here, we treat COVID-19 infection as a hazard and the presence of infectious persons in schools as a threat that might lead to harm.

Having an infected person in a school poses a threat to other individuals, to the operation of the school, which *in extremis* might have to close, and to the wider community. Here, we use the term 'infected school' for a school with one or more infected individuals present. Whether any harm eventuates will depend on many different factors; to be fully pertinent, definitive and auditable, a risk assessment would involve quantification of the probability of this harm occurring.

# 3. Methodology

Our computational platform is the LightTwist UNINET software package (https://lighttwist-software.com/uninet/). UNINET is a standalone uncertainty analysis software with a focus on dependence modelling for high dimensional statistical or probability distributions [6]. The program implements and computes continuous non-parametric Bayesian belief network (BBN) models as the framework for probabilistic modelling. A BBN is a graphical model, which represents joint distributions in an intuitive and efficient way. It encodes the probability density (or mass function) for a set of variables by specifying conditional independence statements and association links in the form of a directed acyclic graph (i.e. network with no closed loops). Some introductory background about the BBN concept is given in electronic supplementary material, Appendix A3 and more technical details on the calculational fundamentals in electronic supplementary material, Appendix A4.

The structure of the model is specified by the analyst to incorporate all essential input variables and conditionalizing factors that are needed to quantify the problem and to enumerate output probability distributions. The approach is general in application and allows traceable and defensible results to be generated very efficiently as UNINET uses unique fast Bayesian updating algorithms. One of the other main advantages of UNINET is that it can handle a very large number of continuous random variables and does not require these to be represented as discretized distributions, as in many BBN programs.

The efficiencies provided by the BBN factorization and the UNINET algorithms allow large sample sets to be generated, and thus determine the distribution tail properties of low probability but high consequence events. This means that tail events can be properly taken into account.

An added benefit is that other factors, elements and complexities can be readily added to the BBN model formulation, for example, by including details of school characteristics or age-dependent transmission rates and, perhaps most pertinently, the BBN calculations can be quickly updated if new or revised data are forthcoming. The model can be applied at different spatial granularities at the

national, regional or local level. As we have found with modelling volcanic eruption risks, this latter capability is invaluable for real-time forecasting applications.

We provide brief information about the UNINET platform in electronic supplementary material, Appendix A2, outlining the essentials of constructing a network as an uncertainty calculator, with a simple specimen network illustrating the form of such stochastic calculations; a BBN analysis summary report from one model run for the present study is reproduced, as an example of model detailing and traceability.

# 4. Data sources for model inputs

We extracted data on primary schools in England, including their size distributions, from publicly accessible UK Government DfE reports and data compilations. Source information for schools' data for 2019 are listed in electronic supplementary material, Appendix A1. Additionally, we used DfE data for daily national school attendance of pupils, teachers, teaching assistants (TAs) and ancillary staff, and for epidemiological data, we used a variety of other sources, listed in electronic supplementary material, Appendix A1.

Basic facts about state-funded primary schools in England in 2019, derived from these sources and used for our modelling, are presented in electronic supplementary material, Appendix A1, table A1.1. Those prevalence data were obtained from infection surveys published by the UK Office of National Statistics (ONS) and incidence data at upper tier local authority (UTLA) level were used as a proxy for regional variations of prevalence; the latter were based on data published by Public Health England (PHE). Population data at UTLA level were obtained from ONS sources.

We divided persons in schools into four Cohorts, as per electronic supplementary material, Appendix A1, table A1.2; these are based on the children's Year groups and also on staff roles. As a simplifying assumption, and in the absence of detailed evidence, our Cohorts were presumed to have different contact characteristics, e.g. with respect to interactions between children, between children and adults and between adults.

The Cohorts chosen were as follows: *Cohort 1* comprised Nursery, Reception and Year 1 children; *Cohort 2* were Years 2–6 children, noting that, for June 2020, Years 2–5 children included those of key workers and from vulnerable environments; *Cohort 3* were classroom teachers and TAs; *Cohort 4* were non-teaching staff such as administrators, cooks, etc., who have more limited contact with children in school (for further details, see electronic supplementary material, Appendix A1, table A1.1).

For June 2020, we were unable to obtain detailed national figures for *Cohorts 3* and *4* because DfE attendance information was not broken down into primary and secondary schools (and may have included some private sector schools). Instead, we drew on information from an elicitation of teachers in 36 primary schools in the Royal Society Schools network, described in a companion study [1]; key aspects of that elicitation study are reported in electronic supplementary material, Appendix A1. These elicited judgements projected an average 65% return of teachers and 58% return of other staff for early June 2020, noting considerable variations between individual schools. Assuming the schools in the Royal Society network are a representative sample of primary schools in England, we estimated national headcounts of 235 000 primary teachers and TAs and 70 000 ancillary staff, after adjusting for the fact that 8% of schools had not re-opened.

Teaching staff on rota and part-time staff further increased the difficulty of determining precise numbers during the first weeks of return so, in the model, we included ±10% uncertainty on these estimates.

# 5. Model structure, scenarios and assumptions

Figure 1 shows the initial exploratory BBN model we established to estimate infection hazard in schools. Input parameters included national/regional prevalence, the susceptibility ratio of children relative to adults (i.e. in the 20 to 60-year age range), and the inventory of children and staff in primary schools.

Outputs included the prevalence in schools, the number of schools with one or more infected person on a given day and how the infection is distributed across different cohorts of persons (e.g. children or adults).

In order to run the BBN model for different circumstances, we adopted a scenario-based approach in which some input parameters are either fixed values or represented by a prescribed probability density function (PDF) reflecting informational uncertainty. The main assumptions for any given scenario are the numbers of pupils, teachers, TAs and ancillary support staff present in schools. Our chosen

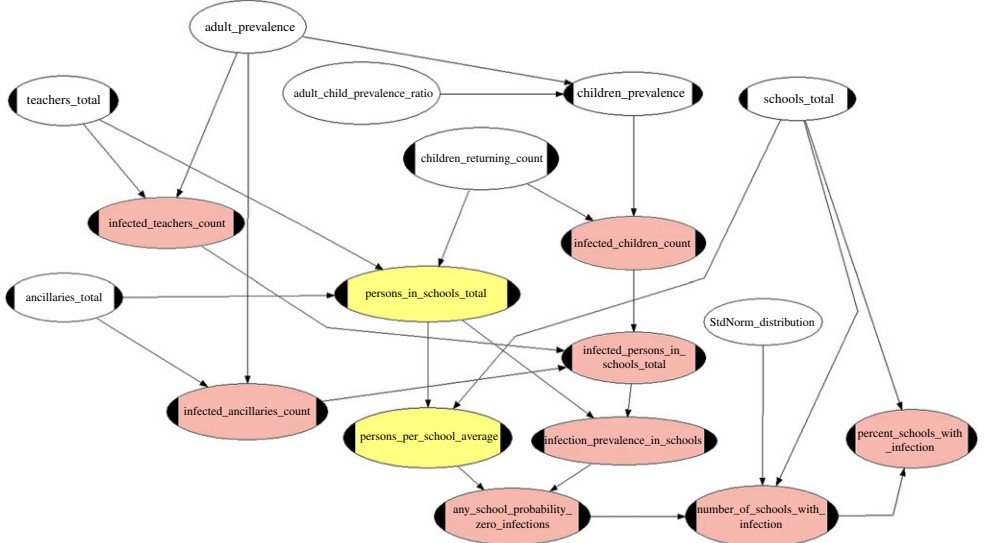

**Figure 1.** Bayesian Belief Network to calculate schools COVID-10 infection hazard with prevalence represented by a single random variable. In order to account for regional variations of prevalence later BBN modelling introduced a range of random variable values for prevalence in different geographical areas. Plain white nodes carry input variable uncertainty distributions, white nodes with handles represent fixed (constant) input values. Yellow nodes are intermediate functional (i.e. calculational) nodes. Required variable probability distributions are computed at the pink nodes; some of these are also intermediate calculation steps, feeding uncertainty distributions into other output nodes.

scenarios, the parameters of which are summarized in electronic supplementary material, Appendix A1, table A1.2, are now described.

*Scenario I* considered the situation on 1 June 2020, the first day of school return for restricted sets of year group children, as stipulated by the UK Government. Numbers of persons attending school were obtained from ONS data, who also published a PDF for adult prevalence on 5 June 2020 (mean value: 1-in-1700); the statistical parameters of the ONS PDF were adopted in the BBN model <*Adult_prevalence*> node to characterize prevalence uncertainty for *Scenario I*.

*Scenario II* assumed attendance of all children in the same selected year groups, adjusted for typical absenteeism (i.e. 4%). Thus, *Scenario II* represents a situation in which there would have been high compliance with the return to school on 1 June 2020.

Looking forward to the full re-opening of schools in September 2020, with all year groups, *Scenario IIIa* assumed the return of all primary school children, adjusted for typical absentee rates. Given substantial forward uncertainty in July 2020 concerning the projected community adult prevalence level in September 2020, rather than simply inflate the ONS PDF by an arbitrary amount we chose to model the potential future prevalence by two marker sub-scenarios: one (*Scenario IIIb*), in which the adult prevalence in the community in September was assumed 4x greater than that given by ONS from their 5 June 2020 survey, and another for a drop in September to one-fourth of the June prevalence (*Scenario IIIc*). We considered that this pair of sub-scenarios represented two contrasting but plausible prospective levels of adult prevalence in September 2020, with associated uncertainties.

The corresponding total numbers of persons in schools for each *Scenario* are given in electronic supplementary material, Appendix A1, table A1.2.

In order to develop the basic infection hazard model to take account of regional variations of prevalence, we used incidence data as a proxy for prevalence because, at the time we developed our model, there was no directly measured prevalence data below the national scale. To first order, prevalence can be related to incidence since, in a steady state, numbers of recovered individuals who are no longer infectious will balance numbers of new infections (while this is a questionable assumption, it allowed immediate model calculations to be tractable). Thus, local or regional prevalence can be scaled from local or regional incidence data: the basis for this conversion is outlined in electronic supplementary material, Appendix A1.

As with all proxies, limitations exist and caveats need to be recognized. Incidence estimated from PCR testing will have been underestimating the true total significantly because not all those infected were tested. At the time, additional testing was focused more on symptomatic people so many

asymptomatic persons were likely to have been missing from published incidence data. While subject to these caveats, use of incidence as a proxy for prevalence enabled us to construct a simplified model of spatial prevalence distributions, scaled by spatial incidence distribution. Electronic supplementary material, Appendix A1, table A1.3 records the data basis for the model of regional prevalence.

The initial challenge for our model was to estimate the base rate number of schools with at least one infectious pre-symptomatic or asymptomatic individual present. We assumed that once symptoms were recognized steps would be taken to isolate the individual from the school and to identify possible contacts. A key issue was that, at the time we performed our analysis, there was mounting evidence that children might be less susceptible to COVID-19 infection than adults. The comprehensive modelling analysis of global data by Viner *et al.* [7] led us to adopt a probability distribution function for the ratio of adult to children's susceptibility with a median of 2.3 and 90% credible interval between 1.3 and 3.8. This range reflected our synthesis of results from other studies in order to characterize, in a single distribution for modelling purposes, the susceptibility ratio between adults and children (e.g. [8–11]). For an urgent risk assessment, we consider this a defensible approach for characterizing the uncertainties associated with some factors; all the distributions used in our model are recorded in electronic supplementary material, Appendix A3.

We also introduced additional variable nodes into the model to allow us to explore the effects of school size distribution on school infection probability. For this purpose, our baseline scenarios assumed all primary schools in England had a mean pupil headcount of 282 and average numbers of classroom and support staff, based on the data in electronic supplementary material, Appendix A1, table A1.1—i.e. the average-sized school. However, the size distribution of primary schools in England, when based on pupil numbers, is wide and polymodal (electronic supplementary material, Appendix A1, figure A1.1). Using this latter size distribution instead of a single 'average school', the influence of school size on potential infection probability can be considered in more detail (§6.3, below).

# 6. Results

In this section, we report our results as values rounded to an appropriate number of significant figures. Given the large uncertainties present in the results, exact values—as provided automatically by the modelling program—can give a misleading impression of precision if such output is reported without qualification. For brevity, we discuss some of the results and compare *Scenarios* in terms of median and mean values. As a basis for enumerating quantitative risk assessments and informing policy decisions, it is essential that parameter uncertainties are properly accommodated in the model and that their influence on outputs are fully articulated; numerical details of all distributions used in the model are recorded in electronic supplementary material, Appendix A3.

## 6.1 Base rate infection hazard levels

We first present results for the distribution of infected schools for the three *Scenarios*, defined above. The numbers of schools with one or more infected persons are listed in table 1, with percentages of all schools in brackets. Corresponding counts of infected children, teachers and support staff, projected to be present in the national primary school system, are given in table 2. As expected, the relative number of schools with infected persons increases in proportion to the increasing number of returning children and teachers.

An example model output distribution is shown in figure 2, showing the distribution of number of schools with at least one infection under *Scenario IIIb* (above). This plot illustrates one BBN result with typical uncertainty spread and some skewness in distribution shape; such long-tailed behaviour, as evinced here, could have implications for policy decision making. This is pertinent because a societal risk assessment that is expressed solely as a central tendency estimate (e.g. as a mean or median, alone) may not be adequate or appropriate for high-consequence risk-informed decision making. Therefore, it is crucial to express such assessment results with a suitable set of confidence levels or exceedance probabilities, such that the decision maker can have a full probabilistic context for appraising the conservatism and tolerability, or otherwise, of the threat.

Inevitably, model parameter uncertainties are large and allow only general inferences. Assuming a fixed prevalence (5 June 2020 ONS data) under conditions extant at that time (*Scenario I*), a few hundred schools were likely to be infected while, with full return in September 2020 (*Scenario IIIa*), the estimated proportion increased by a factor of slightly over 3.6x, in terms of mean numbers of schools involved.

**Table 1.** Infection hazard results: number of state primary schools with one or more infected persons present (with percentage of all primary schools in parentheses)—as at nominal return dates: 1 June 2020 and 1 September 2020.

| model | mean number | 5% quantile | 50% quantile | 95% quantile |
|---|---|---|---|---|
| *Scenario I* Partial 1 June return and 5 June prevalence | 460 (2.7%) | 178 (1.1%) | 406 (2.4%) | 924 (5.5%) |
| *Scenario II* Intended 1 June return and 5 June prevalence | 911 (5.4%) | 336 (2.0%) | 798 (4.8%) | 1873 (11.1%) |
| *Scenario IIIa* 1st Sep return and 5 June prevalence | 1635 (9.8%) | 612 (3.6%) | 1444 (8.6%) | 3310 (19.8%) |
| *Scenario IIIb* 1 Sep return and $4 \times 5$ June prevalence | 5255 (31%) | 2131(12.7%) | 4863 (29.1%) | 9743 (58.3%) |
| *Scenario IIIc* 1 Sep return and $\frac{1}{4} \times 5$ June prevalence | 437 (2.6%) | 162 (0.9%) | 381 (2.3%) | 900 (5.3%) |

**Table 2.** Estimated number of infected persons nationally in primary schools in England, by return/attendance scenarios.

| model | mean number | 5% quantile | 50% quantile | 95% quantile |
|---|---|---|---|---|
| *Scenario I* Partial June return and June prevalence | | | | |
| children | 267 | 88 | 227 | 582 |
| teachers and TAs | 155 | 64 | 139 | 302 |
| ancillary staff | 46 | 19 | 41 | 90 |
| *Scenario II* Intended June return and June prevalence | | | | |
| children | 745 | 246 | 632 | 1624 |
| *Scenario IIIa* Sept return and June prevalence | | | | |
| children | 1417 | 468 | 1203 | 3090 |
| teachers and TAs | 249 | 102 | 223 | 487 |
| ancillary staff | 83 | 34 | 75 | 160 |
| *Scenario IIIb* Sep return and $4\times$ June prevalence | | | | |
| children | 5458 | 1713 | 4572 | 12 188 |
| teachers and TAs | 960 | 372 | 847 | 1193 |
| ancillary staff | 322 | 125 | 284 | 648 |
| *Scenario IIIc* Sep return and $\frac{1}{4}\times$ June prevalence | | | | |
| children | 360 | 121 | 307 | 776 |
| teachers and TAs | 63 | 27 | 57 | 121 |
| ancillary staff | 21 | 9 | 19 | 41 |

The increase in total number of persons attending school between June 2020 (*Scenario I*) and September 2020 (*Scenario III*)—i.e. a factor of 5.6*x*—is greater than the projected increase in the number of infected persons in schools. This is a consequence of the lower susceptibility of children: the bulk of the increase in persons in school, under a September full return (*Scenario III*), can be ascribed to children, as about 65% of adult staff would have already returned at the earlier *Scenario I* stage.

Prevalence of infection in the community will be a major factor in schools' infection hazard. The large uncertainties in the ONS national prevalence infection surveys made it difficult to infer temporal trends. Prevalence fluctuated modestly over the period of the study, declining through June and early July 2020 but increasing during late July and early August. Thus, data as at 5 June 2020 appeared representative through to early August. The effects of prevalence are clear from comparing the projected in-school infection numbers for *Scenarios IIIa, IIIb* and *IIIc* (table 2). If the public maintained the same level of

**Figure 2.** Typical probability density function for projected number of primary schools with infection under *Scenario IIIb* (i.e. September 2020 schools return with 4× June prevalence), using the BBN model of figure 1 (results should be read to nearest whole number—see table 1). Note the heavy upper tail skew to higher values.

adherence to guidance through to September but prevalence dropped to one-fourth of June levels then, in terms of mean values, approximately 2.5% of schools could be expected to have infected persons, compared to 10% at June prevalence levels. However, the emergence of a second wave in autumn causing, say, a factor of 4$x$ increase in prevalence, would have increased the estimated infection rate to about 30% of primary schools.

To illustrate potential implications of the large uncertainties associated with the attendance and prevalence data in terms of an entirely plausible upper-level outcome, we draw attention to the results on table 1. For instance, there could have been about a 1 in 20 chance that the percentage of infected schools under a 4$x$ increase in prevalence (*Scenario IIIb*) might be higher than the 95th percentile estimate: i.e. 58% of all primary schools might have had one or more infected persons upon a full return in September 2020.

Projected prevalence in schools is lower than in the general community because children dominate the school population in terms of numbers. If the general community prevalence had a median value of 1 in 1700 then the corresponding median value of prevalence in schools was estimated as 1 in 2800 for June 2020 return (*Scenario I*) and 1 in 3300 for the base model September 2020 return (*Scenario IIIa*).

A major caveat, addressed below in 6.4, must be that prevalence could vary considerably from region to region, or even from locality to locality; indeed, there might be very localized infection hot spots. For decision making, this is a crucial limitation to simply calculating—and relying on—averaged or overall national values.

## 6.2 What-if? scenario tests

One of the powerful analytic features of the UNINET algorithm underpinning our model is 'what-if?' tests can be implemented directly via its graphic interface, enabling the re-computing model results based on specific parameter values or ranges of values without recourse to additional coding or

**Table 3.** What-if? sensitivity test of selected base model results (i.e. numbers of infected children; infected teachers; infected schools nationally) when BBN figure 1 parameter distribution *Adult_prevalence* is conditionalized equal to or less than its median value or greater than the median.

| | base model | | Adult_prevalence ≤ median | | Adult_prevalence > median | |
|---|---|---|---|---|---|---|
| | mean | s.d. | mean | s.d. | mean | s.d. |
| no. infected children | 5458 | ±3558 | 3402 | ±1838 | 7602 | ±3084 |
| no. infected teachers | 960 | ±511 | 596 | ±240 | 1336 | ±295 |
| no. infected schools | 5255 | ±2350 | 3616 | ±1600 | 7018 | ±1737 |

subroutines. The program can accommodate a single conditionalization that is applied to one variable, or several different conditionalization restraints applied jointly, and simultaneously, to a set of variables in the model. For any parameter, the analyst defines which value or range of values to use for the what-if test of interest.

There can be several reasons for exploring what-if? scenario impacts. For instance, there may be surrogate evidence, say from another country or region, suggesting that a particular parameter likely falls within a certain limited range, when local data are inadequate for constraining that range. The potential implications for our model results—of learning, later, that the uncertainty on the relevant parameter was originally over-stated—can be appraised by conditionalizing on the surrogate range. The UNINET software is ideally suited to this sort of diagnostic what-if? analysis.

With societal risk assessments, decision-makers quite often ask 'what would be the impact on your results if such-and-such a parameter is higher (or lower) than your central value?'. In situations such as this, where new or revised data cannot be waited for, it can be expedient to run some quick, alternative 'what if?' scenario tests for decision makers to consider, especially for parameters or variables that rely on predictions of uncertainty. The risk analyst's role is to provide as much, and only as much, information in the wider context as is advantageous to the decision maker. This said, policy decisions based on such tests remain the responsibility of the decision maker—the analyst should provide neutral and balanced what-if? scenarios.

Here, we illustrate scenario sensitivity testing by conditionalizing one parameter from figure 1, *Adult_prevalence*, two ways, and comparing the re-computed results with the base model results when no conditionalization was applied. Then, we demonstrate what happens if two different variables from figure 1, *Adult_prevalence* and *Adult_child_prevalence_ratio*, are separately and jointly conditionalized, two ways each.

Table 3 summarizes results for: number of infected children, number of infected teachers and number of infected schools. Other model outputs, e.g. for ancillary staff, were available but not repeated here.

Unsurprisingly, conditionalizing the *Adult_prevalence* parameter to restrict it to lower values in its uncertainty range drives projected infection numbers down, and the reverse occurs if a high range is chosen. This said, giving quantitative expression to the size of the effect, made possible by conditionalization testing, can represent valuable informational support to decision making.

Note, too, the consequential reductions in school infection number uncertainties when these conditionalization constraints are applied to the base model—some intrinsic parameter uncertainty is removed by conditionalization.

In the context of COVID-19, these two, very simple, what-if? tests might exemplify, say, how a more stringent policy on social behaviour, or better adherence to guidance, could exert downward pressure on community prevalence and hence on school infection levels, while removing restrictions or increased non-compliance could have the reverse effect. Such hypothetical analyses can help elucidate different decision options.

Next, we illustrate a slightly more convoluted case, with the impacts of conditionalizing two model parameters being considered, jointly. In this analysis, we apply constraints on value ranges for *Adult_prevalence* and for *Adult_child_prevalence_ratio* from figure 1, respectively. Each is constrained to be either above or below its median value, giving four combinations of conditioning, as summarized in table 4 for the same three model outputs reported in table 3, above.

As shown in table 4, from these four-way conditionalizations we can infer that *Adult-Prevalence* has a stronger effect on projected infection numbers than *Adult_child_prevalence_ratio*. While this finding may

**Table 4.** What-if? sensitivity test of selected base model results (i.e. numbers of infected children; infected teachers; infected schools nationally) when figure 1 BBN parameter distributions for *Adult_prevalence* and *Adult_child_prevalence_ratio* are conditionalized equal to or less than their median values or greater than the respective medians.

| | base model | | *Adult_prevalence ≤ median* and *Adult_child_ratio > median* | | *Adult_prevalence > median* and *Adult_child_ratio > median* | |
|---|---|---|---|---|---|---|
| | mean | s.d. | mean | s.d. | mean | s.d. |
| no. infected children | 5458 | ±3558 | 6252 | ±1265 | 8354 | ±2873 |
| no. infected teachers | 960 | ±511 | 807 | ±106 | 1378 | ±299 |
| no. infected schools | 5255 | ±2350 | 5910 | ±792 | 7502 | ±1510 |
| | | | *Adult_prevalence ≤ median* and *Adult_child_ratio ≤ median* | | *Adult_prevalence > median* and *Adult_child_ratio ≤ median* | |
| no. infected children | 5458 | ±3558 | 2745 | ±1219 | 4088 | ±618 |
| no. infected teachers | 960 | ±511 | 547 | ±236 | 1139 | ±169 |
| no. infected schools | 5255 | ±2350 | 3087 | ±1229 | 4759 | ±506 |

be expected in a general sense, the What-if? testing provides a quantitative basis for assessing the relative effects of each, quantified in terms of the related uncertainties.

Note, the multifaceted interplay of uncertainties from just two BBN parameters can give rise to informative results which may be non-obvious, perhaps surprising, even counterintuitive; with such models, subtle aspects of probabilistic reasoning with concordant and discordant data can be present and decision making can benefit from these being elucidated (e.g. [12]).

With complex problems, such diagnostic capabilities are invaluable for appreciating which factors or variables influence outcomes from the model, and by how much. The algorithmic power of UNINET allows conditionalization tests to be applied directly to hundreds of BBN uncertainty nodes, if those uncertainties are represented by conventional statistical distributions with standard functional forms. For other forms of uncertainty distribution, which may not be amenable to parametric PDF shape fitting (e.g. those derived from eliciting expert judgements), UNINET can produce large related sample datasets for analytical post-processing.

We return to these illustrative scenario test findings later, to discuss updating and checking our model findings once more recent data—about infection prevalence levels at the time of the full-scale return to school in autumn 2020—had become available. Then, our early projections could be checked and, if necessary, the model reformulated on the basis of the newer empirical data.

## 6.3. School size effect

Extending the conversation about the base model, the infection hazard for a particular school was presumed to be a function, *inter alia*, of its size. The average sized primary school in England hosts about 280 children. The percentages of all schools infected, reported in table 1, can be taken to apply to a school of this average size, thus giving the probability that such a school will have an infected person on any one day. The mean probability of such an occurrence would have been 1 in 40 for the partial return to school on 1 June 2020 (*Scenario I*) and 1 in 10 for the full return to school in September, with the same community prevalence (*Scenario IIIa*). Using the children count as an approximate proxy for initial school attendance in early June 2020, the mean probability of there being an infected person present in a small rural school with 60 children would have been about 1 in 190 while, for a large urban primary school of 700 children, the corresponding probability would have been 1 in 16.

The number of infected persons in table 2 add up to a slightly larger count (by a few) than the number of schools in table 1. This difference would be due to a small number of schools with two or more infected persons present on the same day. Conditional on at least one infected person being present, the probability of having two or more cases in one school of average size was estimated approximately 0.005 (i.e. 1-in-200), while for three or more it was about 0.00015 (1-in-6667). Note, these probabilities would apply only to infected persons drawn at random from the wider community, assuming average national prevalence.

The estimation of the risk of having multiple infected persons in a school was developed further, below, in relation to spatial prevalence variations (§6.4). Those estimates did not take account of secondary infections occurring due to transmission within the school. Our analysis also assumed that all individuals (pupils, teachers and other staff) would be independent 'samples' in population statistics terms, and therefore their probabilities of being infected uncorrelated. If, however, a school included two or more siblings, then the joint probability of both (or all multiple siblings) being infected together increases relative to the probability of encountering two (or more) unrelated or otherwise unconnected infected members of a population (see electronic supplementary material, Appendix A2).

Between 18 May and 31 July 2020, it was reported that there had been 247 COVID-19 related incidents in schools, of which 116 tested positive [4]. Details about the circumstances for those that tested positive were not accessible, but these data did provide a lower bound on the number of schools with infected persons present. For instance, there may have been a significant number of asymptomatic persons present so it did not follow that having an infected person in a school would lead to an incident culminating in a positive test. Thus, the number of schools estimated to contain an infected person on a given day was much higher than the reported occurrences of positive test cases.

There are many reasons why infection may not be passed on in a school setting, not least because of the stringent risk mitigation measures that were applied at the time in many primary schools; this would have resulted in a much reduced frequency of outbreaks and hence lower disease incidence in schools. In addition, the infectiousness of any particular individual may be low, or they might start feeling ill outside school hours. Infectiousness is thought to be at a maximum just before feeling ill [13,14] and this heightened state might be reached outside school hours (e.g. over a weekend). By contrast, asymptomatic carriers may not be recognized at all. The in-school risk mitigation measures to reduce contacts between people, ensure hygiene and isolate persons who display possible COVID-19 symptoms [1] indicate they were effective in reducing opportunities for infection spread in June–July 2020.

While it might have been tempting to simply scale up linearly from the schools' actual infection outbreak data during *Scenario I* (i.e. June–July 2020) to forecast potential September *Scenario III* incident numbers, using the relative exposed population ratio, we judged it more likely that such scaling would not be linear; in particular, risk mitigation might be less strict and harder to maintain when many more children would be back in school.

The rapid single-day infection snapshot model, described in this paper, can be extended to take into account mitigation measures for infected persons in schools. In addition, it can be implemented as a time-stepping agent-based infection transmission model, accounting for daily temporal variations of numbers presenting in schools as asymptomatic or infected persons; this is on-going work, to be reported elsewhere.

## 6.4. Spatial variations in prevalence

The results in tables 1 and 2, discussed above, assumed a constant prevalence across the UK, but this was clearly not the case at the time, with localized hotspots, such as Leicester, Blackburn and Bradford in July 2020. Regional prevalence levels in early July suggested these varied by a factor greater than $350x$ at UTLA scale (i.e. ONS Upper Tier Local Authority). As a simple illustration, we explored how this variation might affect our projected results. In the BBN, six half-log (i.e. $10^{0.5}$-fold) incidence bins with sampling uncertainties were introduced, with separate contiguous weekly incidence rates, and these were each converted to equivalent prevalence distributions using a generic, uniform scaling factor without allowing for further uncertainties (see electronic supplementary material, table A1.3 in Appendix A1); prevalence values are the inverse of the probability of there being an infected person in a population. For each bin, we calculated from ONS data the total number of persons that tested positive in each bin to estimate a sampling error.

We then modelled regional prevalence differences for a situation equivalent to the full September return (*Scenario IIIa*), with these regional variations corresponding jointly to the overall uniform June prevalence. Table 5 shows prevalence rates partitioned into six regions (*Bins*) with the quantiles of the distributions ascribed to each *Bin*. Also shown are the relative population weightings for each *Bin*. These BBN node histogram distributions are shown in electronic supplementary material, Appendix A1, figure A1.2, with means and standard deviations in the lower panels.

In terms of schools' infection hazard results, table 6 shows the statistical characteristics of our estimated number of state primary schools in England with one or more infected persons for the September full return *Scenario IIIa* base model versus the multi-prevalence model, outlined above (see also electronic supplementary material, Appendix A1, figure A1.2); the equivalent percentages of all

**Table 5.** Model spatial distribution of prevalence per 100 000 persons, dividing England into six contiguous half-log prevalence regions with weightings reflecting the relative population sizes of these regions. The net mean prevalence in the BBN, in probability terms, is slightly higher than the corresponding overall mean from the tabulated data (1-in-1645 versus 1-in-1700). This is likely due to slight misfits when converting *Bin* raw data into separate statistical distributions.

| prevalence | data mean | BBN mean | 5th percentile | 50th percentile | 95th percentile | population weighting |
|---|---|---|---|---|---|---|
| Bin 1 | 4 | 3 | 1 | 4 | 6 | 0.07 |
| Bin 2 | 12 | 12 | 7 | 11 | 18 | 0.19 |
| Bin 3 | 34 | 33 | 19 | 32 | 52 | 0.42 |
| Bin 4 | 95 | 95 | 57 | 91 | 145 | 0.27 |
| Bin 5 | 259 | 256 | 159 | 247 | 385 | 0.04 |
| Bin 6 | 841 | 841 | 546 | 841 | 1140 | 0.01 |
| overall means | 59 | 61 | | | | |
| probability 1-in | 1700 | 1645 | | | | |

**Table 6.** Summary of infection hazard results showing: the number of state primary schools with one or more infected persons present (with percentage of all schools in parentheses). Results for *Scenario IIIa* base model are compared with results from the multi-prevalence model (row 2 'Spatial prevalence') and then from the multi-prevalence model including school size distribution (row 3). (See text and electronic supplementary material, Appendix A1, figure A1.1).

| | mean | s.d. | 5% quantile | 50% quantile | 95% quantile |
|---|---|---|---|---|---|
| *Scenario IIIa* base | | | | | |
| 1 Sep return and 5 June prevalence | 1635 (9.8%) | 732 | 612 (3.6%) | 1444 (8.6%) | 3310 (19.8%) |
| spatial prevalence | 1458 (8.7%) | 1873 | 122 (0.7%) | 888 (5.3%) | 4344 (25.9%) |
| school size distribution | 1405 (8.4%) | 1910 | 59 (0.35%) | 754 (4.5%) | 4910 (29.3%) |

schools are shown in parentheses. The prevalence model with spatial variation predicted a slightly reduced number of infected schools compared with the uniform prevalence model. Estimated numbers of infected persons in primary schools, by pupil, teacher or ancillary staff, for the September return *Scenario IIIa* base model versus the multi-prevalence model, are shown in table 7. In terms of mean values, the numbers are almost identical. However, the variances on the number of schools and infected numbers of persons in the spatial model are increased relative to the base model, with more marked skewness reflected in the lower median values compared to the uniform prevalence model.

To model the distribution of infected schools in areas of different prevalence, we individually modelled each bin with school numbers proportioned to bin population. Table 8 shows the results comparing the bins by number of infected schools per 100 000 people. One major and unsurprising consequence of the spatial variation of prevalence is that the number of schools with infected individuals is strictly proportional to prevalence. The corollary of this is a lower mean number of schools with infection (and lower numbers of infected pupils, teachers and ancillaries) in areas of lower prevalence. Figure 3 displays the results as per cent of infected primary schools that would exceed a given prevalence value. For this illustrative scenario, 82% of infected schools would be in areas of above-average prevalence.

One of the consequences of high prevalence in an area is to concentrate, in that area, the number of schools that have two more infected persons on the same day. An approximate measure of the number of schools that have multiple infected persons is the difference between the number of schools with one or more infected persons and the corresponding infected person numbers: the excess infected cases indicate some schools in a high prevalence area can expect to have two or more cases at any one time. For the September 2020 uniform prevalence model (i.e. with 5 June 2020 prevalence: *Scenario IIIa*), the difference between the number of schools with an infected person and the number of infected persons was 114, but for the spatial prevalence model, the difference increased to 288.

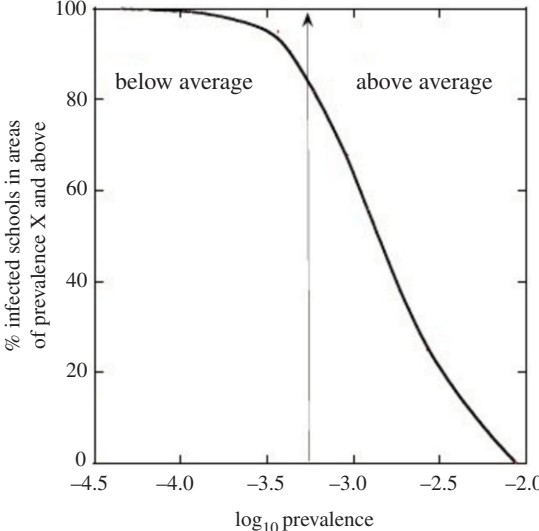

**Figure 3.** Survivor function plot showing percentage of infected schools in areas of prevalence X and above. The percentage of infected schools that occur at or above a fixed value of prevalence can be read off the curve. The model is *Scenario IIIa* with regional prevalence variation. The central vertical line is the national average prevalence assumed in the model. Eighty-two percentage of infected schools are located in areas with prevalence above the national average.

**Table 7.** Estimated number of infected persons in primary schools: *Scenario IIIa* base model versus multi-prevalence model (see text and electronic supplementary material, Appendix A1, figure A1.1).

|  | number | s.d. | 5% quantile | 50% quantile | 95% quantile |
|---|---|---|---|---|---|
| *Scenario IIIa* base Sep return and June prevalence | | | | | |
| children | 1417 | 883 | 468 | 1203 | 3090 |
| teachers and TAs | 249 | 125 | 102 | 223 | 487 |
| ancillary staff | 83 | 42 | 34 | 75 | 160 |
| spatial prevalence | | | | | |
| children | 1415 | 2699 | 95 | 726 | 4167 |
| teachers and TAs | 248 | 443 | 19 | 136 | 652 |
| ancillary staff | 83 | 149 | 6 | 46 | 219 |

**Table 8.** Distributions for estimated average number of schools with one or more infected person as a function of spatial prevalence levels; calculations are based on the assumption that the total numbers of schools per prevalence bin are proportional to the corresponding population size. Different school size profiles within prevalence bins are not accounted for here, so the tabulated numbers of infected schools relate to an average sized school.

| prevalence | mean no. infected schools | s.d. | 5th percentile | 50th percentile | 95th percentile | no. infected schools per 100 000 population |
|---|---|---|---|---|---|---|
| Bin 1 | 8 | ±3 | 5 | 8 | 13 | 0.2 |
| Bin 2 | 55 | ±15 | 35 | 53 | 83 | 0.5 |
| Bin 3 | 377 | ±98 | 244 | 363 | 559 | 1.6 |
| Bin 4 | 592 | ±147 | 391 | 572 | 862 | 3.9 |
| Bin 5 | 221 | ±47 | 153 | 216 | 306 | 9.7 |
| Bin 6 | 121 | ±15 | 96 | 121 | 145 | 34 |
| overall total | 1374 | | | | | |

This result is significant in that it predicted there would be a factor of almost $3x$ increase in the risk of an outbreak of multiple infected persons in some schools.

## 6.5. Effects of school size and multiple infections

Schools vary significantly in size (electronic supplementary material, Appendix A1, figure A1.1). We expected that the probability of encountering one or more infected persons in a school would depend on size. We, therefore, ran a model which sampled from the national distribution of school sizes, rather than assuming a mean school size. The model chosen was *Scenario IIIa* with spatial prevalence variation. The results are shown in table 6. Including variation in school size reduces the anticipated number of infected schools from 1458 to 1405, while the uncertainty range increases slightly. The decrease is because of the higher probability of an infected person being encountered in a larger school. Since there is no change in the overall number of infected persons, the implication of a smaller mean number of schools with infection is that there would be more cases of multiple infections in some schools.

The discussion, so far, has focused on estimating the probabilities of encountering one or more infected persons in a school. Because there could be implications for transmission risk, our analysis was extended to enumerate probabilities for 2, 3, 4, …, N infected persons per school. For the September full return model (*Scenario IIIa*)—including the school size distribution and regional prevalence variations—our model computed mean probabilities, as a function of N, as follows:

$$N = 0 \; \text{Prob} = 0.916$$
$$N = 1 \; \text{Prob} = 0.072$$
$$N = 2 \; \text{Prob} = 0.009$$
$$N = 3 \; \text{Prob} = 0.002$$
$$N = 4 + \text{Prob} = 0.001$$

By way of illustration, we contrast a small school (50 pupils; 10 teachers; 10 other staff), in an area with the second-lowest spatial mean prevalence (table 5, Bin 2), with a large school (900 pupils; 100 teachers; 100 other staff) in the second-highest prevalence area (table 5, Bin 5). The estimated probability of one infected person (pupil or adult) in the small school was $4.8 \times 10^{-3}$; for two or more the probability would be about $1.1 \times 10^{-5}$. By contrast, for the large school the probabilities of N infected persons were

$$N = 0 \; \text{Prob} = 0.26$$
$$N = 1 \; \text{Prob} = 0.35$$
$$N = 2 \; \text{Prob} = 0.24$$
$$N = 3 \; \text{Prob} = 0.11$$
$$N = 4 \; \text{Prob} = 0.04$$
$$N = 5 + \text{Prob} = 0.01$$

Thus, for a very big primary school in September 2020 in an area with high prevalence (table 5, Bin 5), we calculated there was a 40% chance of having two or more infected persons and about a 1% chance of encountering five or more infected persons in the school at any one time.

## 6.6. Comparison with observations

Our model for the 1 June 2020 partial return (*Scenario Ia*) gave a mean of 460 (90% credible range 178.924) infected schools on any one day. Taking account of regional prevalence variations and school size distributions this expected number would be reduced by about 15%, to 390 schools (table 6).

With time, the numbers of schools in which an infected person has been present will slowly grow, implying the number of infected schools would be much higher after a few weeks. A quite complicated calculation would be needed to infer what numbers of primary schools could have been infected in June and July. For instance, the calculation would need to consider both temporal and spatial variations of prevalence and also enumerate variations of how long an individual remains infectious. Instead, a very approximate simplified calculation could give a feel for the likely number. If infectiousness typically lasts a week [13], then for fixed prevalence and a daily rate of 55 new infected schools per day (e.g. per table 8), the total number of infected schools in July 2020 would have been about 1925.

Between 18 May and 31 July 2020, 247 COVID-19 related incidents were reported in primary schools in England, of which 116 tested positive [2] (see also [4]). Even allowing for the large uncertainties, these numbers are below our estimated number of infected schools. However, it seems likely that many infections were not detected because those persons were asymptomatic or developed unreported

symptoms outside school hours. The low number could also have reflected incomplete testing coverage at the time. Further, the probability of onward transmission to another person in the school could have been substantially less than unity.

We have shown elsewhere [1] (see also [15]) that, in June–July 2020, close contacts between persons in primary schools were reduced by between 45% and 75%, compared to normal pre-COVID times while, in addition, extra risk mitigation measures would have reduced transmission risk still further.

# 7. Checking the model

## 7.1. Reviewing scenario projections

As just discussed, our stochastic uncertainty model for infections in primary schools was used to generate predictive estimates of COVID-19 infections in schools as at 1 September 2020, marking a nominal start of the 2020/2021 academic year. To illustrate the flexibility of the BBN model, we ran it with three alternative scenarios for credible future levels of community prevalence in September 2020, as described above (§5). Subsequently, it was possible to test the September 2020 scenario projections against actual attendance records, and to update the model configuration with revised data on the estimated prevalence in early September (table 5).

To check the BBN projections, we use prevalence estimates from an ONS sampling survey reported on 10 September 2020 (n.b. ONS methodology for estimating population prevalence changed on 6 July 2020, noting 'Headline figures (from 6 July) not comparable to previous estimates'.). The ONS survey reported a population infection prevalence of 0.16% with 95% credible interval 0.13–0.19%; this adjusted version of the September base scenario we label *Scenario III\** in table 9. This prevalence mean value is approximately 2.5x the prevalence of 5 June 2020, but lower than the mean prevalence assumed for our projections with 4x June prevalence under *Scenario IIIb*. When updating the model to match *Scenario III\** prevalence, this difference would be expected to reduce the numbers of infected individuals when compared with the 4x prevalence *Scenario IIIb*; indeed, this is observed in the number of infected teachers and TAs and ancillary staff which are substantially lowered.

However, under *Scenario III\** we also update the *Adult_child_prevalence_ratio*, based on a breakdown of the ONS population prevalence into age-group categories. For 10 September 2020, ONS estimated the prevalence for Age 2 to School Year 6 children at 0.16% (95% CI 0.09–0.27%) which compares to 0.17% (0.11–0.27%) for Age 25–34, 0.09% (0.06–0.14%) for Age 35–49, 0.08% (0.05—0.11%) for Age 50–69 and 0.06% (0.03–0.10%) for Age 70+. Similarly, the elevated prevalence for primary school ages relative to older age groups was a persistent characteristic of the ONS estimates throughout September. However, there was a more complex pattern of prevalence among older children, with the prevalence of secondary school children lower than that for primary ages until 13 September, but thereafter it rose rapidly. Prevalence among young adults (school Year 12 to Age 24) was substantially higher and increased rapidly through September.

For evaluating *Scenario III\** in the revised model, we chose to amend the *Adult_child_prevalence_ratio* to reflect these observations, adopting a lognormal distribution with mean 1.3 and standard deviation 0.25 (0.9–1.28–1.75; 5%ile–50%ile–95%ile) to represent uncertainty. This results in relatively higher infection rates in children, so that *Scenario III\** predicts a larger number of pupil infections than our original 4x prevalence *Scenario IIIb*, notwithstanding the lower community population prevalence indicated by the data reported by ONS.

The DfE estimated that on 10 September 2020, 99.9% of state schools were (at least partially) open and that 88.3% of children attended school (Week 37 data). Reports of numbers of schools with children self-isolating due to COVID-19 contact within the school were not available prior to Week 42 (12 October); subsequently, numbers were estimated by DfE at 4400 on 12th, 4500 on 13th, 4600 on 15th and 4500 on 16th October. Note, these numbers were for all state schools, with primary schools comprising 68.9% of the total.

While there may be increased levels of symptomatic infections in older children and greater opportunity for contacts in secondary schools, on the basis of the DfE October 2020 figures and primary schools being 68.9% of the total, we infer that roughly 3000 primary schools could have had self-isolating pupils in the latter half of September 2020. This number is lower than the projected median on tables 10 and 11 for 4x June prevalence *Scenario IIIb* (i.e. 4863 schools), but well within the corresponding 90% CI estimate.

**Table 9.** Summary of ex ante and *post hoc* modelling of infection hazard in primary schools in September 2020. Summary of uncertain network results. Scenarios differ in the major uncertainties: <*Adult_prevalence*> and <*Adult_child_prevalence_ratio*>. The mean number, standard deviations and the 5%, 50% (median) and 95% quantiles of the distributions for children, teachers and TAs, and ancillary staff are shown.

| | Adult_prevalence (mean, s.d.) | Adult_child_prevalence_ratio (mean, s.d.) | numbers of infected persons | | | | number of schools with one of more infected persons present | | |
| --- | --- | --- | --- | --- | --- | --- | --- | --- | --- |
| | | | group | number | s.d. | 5–50–95% quantiles | mean | s.d. | 5–50–95% quantiles |
| *Scenario IIIa* (Sep 2020 return all pupils) | (0.066%, 0.033%) prevalence of 5 June | (2.34, 0.777) | children | 1417 (0.03%) | 883 | 468–1203–3090 | 1635 (9.8%) | 732 | 612–1444–3310 (3.6–8.6–19.8%) |
| | | | teachers and TAs | 249 | 125 | 102–223–487 | | | |
| | | | ancillary staff | 83 | 42 | 34–75–160 | | | |
| *Scenario IIIb* (Sep 2020 return all pupils) | (0.266%, 0.033%) 4× prevalence of 5 June | (2.34, 0.777) | children | 5458 (0.12%) | 3558 | 1712–4572–12188 | 5255 (31%) | 2350 | 2131–4863–9743 (12.7–29.1–58.3%) |
| | | | teachers and TAs | 960 | 511 | 372–847–193 | | | |
| | | | ancillary staff | 322 | 171 | 125–284–648 | | | |
| *Scenario IIIc* (Sep 2020 return all pupils) | (0.016%, 0.033%) ¼× prevalence of 5 June | (2.34, 0.777) | children | 360 (0.008%) | 219 | 121–307–776 | 437 (2.6%) | 243 | 162–381–900 (0.9–2.3–5.3%) |
| | | | teachers and TAs | 63 | 31 | 27–57–121 | | | |
| | | | ancillary staff | 21 | 10 | 9–19–41 | | | |
| *Scenario III** (Sep 2020 return all pupils) | (0.16%, 0.02%) prevalence per ONS survey 10 Sep 2020 | (1.3, 0.25) | children | 5791 (0.13%) | 1336 | 3879–5643–8206 | 5198 (31%) | 921 | 3801–5133–6817 (22.7–30.7–40.8%) |
| | | | teachers and TAs | 376 | 60 | 284–372–482 | | | |
| | | | ancillary staff | 112 | 18 | 84–111–143 | | | |

**Table 10.** Sensitivity analysis for alternative incidence-to-prevalence conversion factor—effect on estimated numbers of infected pupils and infected teachers nationally in primary schools in England, for return to school in September 2020—*Scenario IIIA* (see also §6.1 and table 2). n.b. while modelling results are reported to nearest whole number, such precision is not claimed for these indicative projections.

| model | mean number | 5% quantile | 50% quantile | 95% quantile |
|---|---|---|---|---|
| *Scenario IIIa* basis: Sep 2020 return and inferred June 2020 prevalence (per table 2); incidence-to-prevalence factor 6× | | | | |
| children | 1417 | 468 | 1203 | 3090 |
| teachers and TAs | 249 | 102 | 223 | 487 |
| *Scenario IIIa* with incidence-to-prevalence factor 4× | | | | |
| children | 944 | 356 | 828 | 1927 |
| teachers and TAs | 166 | 79 | 153 | 295 |
| *Scenario IIIa* with incidence-to-prevalence factor 8× | | | | |
| children | 1888 | 711 | 1655 | 3855 |
| teachers and TAs | 331 | 159 | 306 | 589 |

**Table 11.** Sensitivity analysis of table 2 *Scenario IIIa* projections for in-school infection numbers in September 2020 if, in the interim, prevalence departed from the June 2020 basis model steady-state assumption. n.b. computed results are shown to nearest whole number, but such precision is not claimed for these indicative projections.

| model | mean number | 5% quantile | 50% quantile | 95% quantile |
|---|---|---|---|---|
| *Scenario IIIa* basis: Sep 2020 return and inferred June 2020 prevalence (per table 2); incidence-to-prevalence factor 6× | | | | |
| children | 1417 | 468 | 1203 | 3090 |
| teachers and TAs | 249 | 102 | 223 | 487 |
| *Scenario IIIa* with prevalence increasing within 50th to 95th percentile uncertainty range | | | | |
| children | 1730 | 890 | 1609 | 2990 |
| teachers and TAs | 304 | 236 | 292 | 411 |
| *Scenario IIIa* with prevalence decreasing with 5th to 50th percentile range | | | | |
| children | 1002 | 516 | 940 | 1706 |
| teachers and TAs | 176 | 129 | 177 | 218 |

For instance, our model included two levels of COVID-19 community prevalence (i.e. 0.066% as at 1 June 2020; and a hypothetical September prevalence $4x$ higher, i.e. 0.27%) and, for these two scenarios, IIIa and IIIb, projected mean numbers of infected primary schools were 1635 and 5255, respectively (table 9). ONS data indicate the prevalence in primary school-age children in September–October 2020 was about 0.16%, i.e. roughly midway between our mid- and upper prevalence scenario assumptions. Given our model indicated single cases in the great majority of those schools with any infected person, jointly our pair of original scenario projections encompassed the eventual reported count of roughly 3000 primary schools with self-isolating pupils in late September 2020.

The discussion of conditionalized example 'What-if' analyses, presented in §6.2 above, included two cases where the mean number of infected schools were very close to 3000. One represented a single conditionalization, with *Adult_prevalence* samples constrained to values in the lower 50% of the $4x$ prevalence distribution; this indicated an expected mean of 3616 (±1600) infected schools in September 2020.

When that conditionalization was coupled jointly with the *Adult_child_prevalence_ratio* being also restricted to the lower half of its distribution, the net effect was a projected mean of 3087 (±1229) infected schools—i.e. very close to the likely count of circa 3000 schools in late September 2020, indicated by DfE estimates for early October 2020.

There is evidence from demographic epidemiological studies (e.g. [16]) that infected young children are more likely to present asymptomatically than older children and adults. While, on one hand, our BBN model may have not allowed adequately for the number of asymptomatic pupils who did not isolate, on the other hand, it could have over-estimated the total number of infected children in primary schools.

We note also that the updated prevalence model *Scenario III\**—which incorporated distribution revisions for *Adult_prevalence* and for *Adult_child_prevalence_ratio*—indicated an expected mean number of 5198 infected schools in September 2020 and that the corresponding 3000 estimate from DfE falls a little below the 5th percentile value projected by our original model (i.e. below 3801 schools, per table 9).

The best we can say at this stage is that uncertainties in our BBN model, in relation to these two factors (*Adult_prevalence* and *Adult_child_prevalence_ratio*), likely cancelled each other out to some extent. This said, *post hoc* we found that the teachers' judgements [1] that informed our BBN model parameters for the initial partial return to school in early June 2020, and the projections we generated in May–June, were themselves realistic and reasonable when it came to numbers of schools with infections in September–October, following the start of the 2020/2021 academic year.

Detailed model diagnostic examination—possible with the UNINET package—can elucidate the influence of key factors and of other parameters on our initial infection estimates; next, we demonstrate this capability with a pair of sensitivity analyses.

## 7.2. Incidence-to-prevalence conversion factor—sensitivity analysis

As discussed in §4, we examined regional incidence data—available from June 2020—and used these as ergodic analogues for prevalence. On this basis, we derived a simple relation between incidence and prevalence, leading to the assumption that the latter can be considered approximately proportional to $6x$ incidence under steady-state infection-response conditions at the time (per electronic supplementary material, Appendix A1). In our stochastic model, this incidence multiplier is represented by an empirical two-parameter Lognormal uncertainty distribution with scale = 6 and shape determined by combining sets of geographical-referenced incidence data (electronic supplementary material, Appendix A1).

In table 10, we examine the effect on our estimates of national numbers of infected pupils and infected teachers under *Scenario IIIa* (see also §6.1 and table 2) by re-running our model with Lognormal prevalence scale changed to 4 or 8, instead of 6, while retaining the same shape factor.

Overall, the means of these sensitivity results directly reflect the corresponding changes in prevalence scaling, as might be expected in an essentially linear model. The quantiles of the associated stochastic distributions also reflect the scaling factors, albeit with minor variations likely due to stochastic sampling effects in the different computational runs with UNINET.

## 7.3. Incidence/prevalence 'steady state' assumption—sensitivity analysis

In §4, the basic assumption underlying the incidence-to-prevalence factor in our model, just discussed, was that COVID-19 incidence and prevalence could be presumed to equilibrate into steady-state statistical processes at national and regional scales and that this would remain stable from June 2020 (i.e. the incidence data then available) through to return-to-school in September 2020. The possibility existed, however, that infection processes or incidence rate itself could evince trend changes with time, such that our assumption of a constant incidence-to-prevalence factor would be questionable. Therefore, it is reasonable to ask about potential effects on our results if the steady-state assumption were not fully valid.

Bearing in mind our model simply presents a nominal single-day future projection of in-school infection numbers, here we take advantage of UNINET's conditionalized re-sampling capabilities to test the impact on our projections of upward or downward systematic changes in prevalence over the forward period from June to September 2020, again using *Scenario IIIa* results from table 2 as reference. To demonstrate the sorts of sensitivity analysis that can be undertaken, we re-run the projection model with two conditionalizations applied: (i) prevalence from June 2020 might have followed an upward trajectory, falling somewhere within its 50th to 95th percentile uncertainty range; and (ii) a downward trend in prevalence might have occurred, falling within its 5th to 95th percentile uncertainty range. Full stochastic re-sampling leads to the results summarized in table 11.

With the modelled general upward trend in prevalence, the infection number distributions for children and teachers and TAs are both skewed with longer upper tails (table 11). For the conditional

numbers under a general downward trend in prevalence, the children's distribution is still marginally skewed with a slightly longer upper tail, whereas the teachers and TAs distribution support is reversed, more toward smaller numbers.

These two sensitivity analyses illustrate how our model can be interrogated for the impacts of different circumstances, conditions or parameterizations. While the number of alternative scenarios that can be quantified probabilistically in this way is almost limitless, such sensitivity tests are best tailored to the policy concerns of the problem owner or decision maker.

# 8. Further work

One factor not fully explored in our initial study in June–July 2020 was the size distribution of primary schools. We assumed that the national size distribution applies everywhere, but this size distribution is likely to vary at UTLA scale across the country, depending on the character and make-up of the local community Thus, our results can be anticipated to change if recalculated with greater granularity at Lower Tier or school catchment scale, for example. A city like Leicester (a single UTLA) could, itself, be modelled at much finer local scale to understand which individual schools within that conurbation would have higher COVID-19 infection hazard. While our punctual model captured the first order picture, it did not have the necessary spatial resolution to explore infection hazard at local levels. Postcode-based incidence and prevalence data would allow modelling to address risks at an individual school's catchment scale.

For incidence parameter uncertainty characterization, we adopted the assumption of lognormal distributions for the different partitions of the UTLA geographical-referenced data. As always, there can be a number of reasons for settling on one preferred distribution type, or another; in many disciplines, when treating positive real variable data, a lognormal process is regarded as the statistical realization of the multiplicative product of several factors, which is likely to be true with incidence/prevalence data. The lognormal distribution has the advantage of common statistical usage and understanding, hence we adopted it here in our initial 'scoping' analyses. Other distributions (e.g. Weibull; gamma) may represent more appropriate functional forms for incidence/prevalence data when aspects like goodness-of-fit are considered. We intend to explore alternatives objectively in follow-on work, paying attention to upper tail fitting because, in a safety-critical risk assessment, it is there that low probability, but potentially disastrous outcomes, may reside. The likelihood of occurrence of such threats warrants enumeration as reliably as possible, in the face of the many intrinsic uncertainties.

Our BBN model, as set up, can also be configured straightforwardly to estimate infection risk in schools of specific sizes (i.e. small, medium or large) or for individual schools, of given size, in areas with known, specific prevalence rates. The model can be operationalized at national and local levels to give near real-time infection hazard informed by weekly changes of incidence and prevalence. The model is thus suitable for decision support with regard to management strategies at school, city, local authority and national scales, and for informing other research programmes, e.g. infection testing in schools.

This discussion presents the findings of an initial first stage analysis, executed in some haste in June–July 2020 and posted as a *medRxiv* preprint [15]. Our school infection rate estimates did not include the possibility of individual schools having sibling pupils who may be simultaneously infectious; these would need to be modelled as non-independent cases, i.e. correlated samples, within a school population. In similar vein, there could be schools with twins or triplets in the same class; this, too, would affect potential infection and onward transmission rates. An initial scoping exercise, however, suggests that co-sibling infection rates are low and, as the proportion of a school pupil population that is represented by siblings is also small, the influence of children living at the same home address is likely to be inconsequential for estimating school population infection rates.

Our current BBN model does not provide a direct enumeration of outbreak risk in schools, although we did surmise the infection hazard is strongly correlated with outbreak risk. An adjunct transmission model could be added to our infection hazard BBN to characterize the probability of occurrence of an outbreak of detected infections, given the presence of one or more infected individuals in a school.

We noted that our initial model estimated a mean of a few hundred infected primary schools per day in England during June, whereas reported incidents, *with associated positive tests*, amounted to much less than this (about 116 cases). While hypothesizing that actual infections were likely to have been undetected or under-reported and that evidence for active in-school infections by transmission was probably suppressed by rigorous risk mitigation measures implemented in the schools, more work is

needed to understand which factors engendered the differences that emerged between projected and reported rates of infection.

This said, retrospective what-if? scenario tests with our BBN model demonstrated that, if uncertain parameters could be rationally constrained, then skilful projections could be derived for COVID-19 infection levels for partial re-opening of primary schools in June and for their pre-planned full return in September 2020.

# 9. Informing policy decisions

The main policy implications of our preliminary infection hazard results for June 2020 were that in England a few hundred primary schools were expected to have infected individuals in early June 2020, just after re-opening. With a partial return to school, social distancing measures were relatively easy to comply with. On this basis, for a full return to school in September, similar numbers of schools were expected to have infected persons present on one day, but only provided that prevalence falls to about one-fourth of the June 2020 level; on the other hand, numbers could be expected to more than triple if prevalence stayed at those levels.

Our model indicated that, if a national scale second wave came with, for example, community prevalence increasing by a factor of $4x$ compared to June 2020, several thousand schools might have infected persons present. This despite the fact that net prevalence in schools is likely to be much lower than in the general community because of the high proportion of persons in schools being children with lower infection susceptibilities. While infection hazard is proportional to school size, it also depends on the relative proportion of children and adults in the school. Social distancing measures would be much harder to implement with a full return to school.

A variant of our model, which explored the effects of spatial prevalence variations, found that schools with infected persons are preferentially concentrated in high prevalence communities. By the same token, there was an overall slight decrease in the projected number of infected schools nationally. However, in high prevalence areas, there would be an approximately three-fold increase in the likelihood of some schools having more than one infected person attending school at any one time, even without considering the effect of siblings and transmission leading to secondary infections. Unsurprisingly, we find infections will tend to be concentrated in larger schools.

When regional prevalence variations are introduced into the Scenario IIIa model (i.e. September full return to school with average June community prevalence overall), the number of infected schools per 100 000 people varies by a factor of about $170x$ between the lowest and highest prevalence areas in England at the UTLA scale, at the time (table 8). This large variation suggests a strong policy focus on mitigating risk in high prevalence areas. In our illustrative model, 82% of infected schools are located in areas with prevalence above the national average. One option for the government or for a local authority is to consider stricter risk management regimes in schools where and when local incidence rates exceed some threshold. To set such a threshold on a reasoned and defensible basis, requires a formal quantitative risk assessment, with uncertainties properly incorporated.

Our results were also consistent with the effectiveness of risk mitigation measures taken in schools to reduce transmission. The number of incidents in schools with positive tests was less than the number of expected schools with infected persons present by at least an order of magnitude. Unfortunately, interpretation of the difference is confounded by other factors (e.g. completeness of testing and role of asymptomatics). Nonetheless, the reduction of contacts within primary schools of between 45% and 75% ([1]; fig. 6) and instigation of other safety measure (e.g. handwashing and fastidious cleaning) supports our inference that risk mitigation was a significant factor in June–July 2020.

These numerical projections and findings represent a contribution to the on-going UK policy-making debate about the effect of school closures on COVID-19 disease, in terms of predictions of infection levels, illnesses and mortality (e.g. [17] and responses therein).

One other thing this rapid study demonstrated is the diagnostic power of the UNINET stochastic uncertainty program for calculating, and understanding, predictive aspects of complex, convoluted models with significant elements of uncertainty. This capability is applicable to any number of modelling issues associated with COVID-19 projections where uncertainty is endemic, and contrasts with other analyses where such challenges are evaded for want of recourse to this numerical uncertainty methodology (e.g. paras 2.16 and 3.20 in [18]).

The two sensitivity analyses, described above (§ 7.2 and 7.3), serve to illustrate how our model can be interrogated for the impacts of different circumstances, conditions or parameterizations. Such sensitivity

tests can be customized exactly to the concerns of the problem owner or decision maker for policy setting in any future societal infection wave or new outbreak. In this regard, an infection transmission algorithm and model diagnostic capabilities have been extended to accommodate the dynamics of new variant coronavirus infections in schools, allowing efficacies of alternative mitigation measures to be gauged objectively.

Data accessibility. Data are reported in the main text, in the electronic supplementary material or are available from the sources cited in the text or in the electronic supplementary material. Enquiries about data should be directed to the first author. The UNINET software package link is: https://lighttwist-software.com/uninet/.
    The data are provided in the electronic supplementary material [19].
Authors' contributions. R.S.J.S. and W.P.A. designed and coordinated the project and led the writing of the paper. R.M.C. and M.J.W. supported data acquisition and processing, BBN modelling and interpretation of results. J.H.S. supported the project with data acquisition and collation. Colleagues in the CoMMinS Project provided epidemiological advice and guidance. All named authors contributed to the discussion and to finalizing the manuscript.
Competing interests. The authors declare no competing interests.
Funding. There was no direct funding for this study but the support of the Royal Society education policy unit and the Royal Society RAMP initiative for COVID-19 are acknowledged. The 'COVID-19 Mapping and Mitigation in Schools' (CoMMinS) project was supported by the Medical Research Council (grant no. MR/V0285545/1) and hosted within the UK MRC Integrative Epidemiology Unit at the University of Bristol (grant no. MC_UU_00011/5).
Acknowledgements. We thank Leon Danon and Jonty Rougier for commenting on an earlier version of this paper. We are grateful to the RSOS editors and two anonymous reviewers for many constructive and insightful suggestions that helped significantly improve this contribution.

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
