## [Peer Review File · Royal Society Open Science]

Review History

RSOS-202218.R0 (Original submission)

Review form: Reviewer 1

Is the manuscript scientifically sound in its present form?

Yes

Are the interpretations and conclusions justified by the results?

Yes

Is the language acceptable?

Yes

Do you have any ethical concerns with this paper?

No

Have you any concerns about statistical analyses in this paper?

No

Recommendation?

Accept with minor revision (please list in comments)

Comments to the Author(s)

See attached pdf file (Appendix A).

Review form: Reviewer 2

Is the manuscript scientifically sound in its present form?

No

Are the interpretations and conclusions justified by the results?

No

Is the language acceptable?

Yes

Do you have any ethical concerns with this paper?

No

Have you any concerns about statistical analyses in this paper?

No

Recommendation?

Reject

Comments to the Author(s)

Comments attached (see Appendix B).

Decision letter (RSOS-202218.R0)

Dear Dr Aspinall,

The Editors assigned to your paper RSOS-202218 "Quantifying risk levels in Primary Schools in England from COVID-19 infection hazard" have now received comments from reviewers and would like you to revise the paper in accordance with the reviewer comments and any comments from the Editors. Please note this decision does not guarantee eventual acceptance.

Please submit your revised manuscript and required files (see below) no later than 21 days from today's (ie 03-Jun-2021) date. Note: the ScholarOne system will 'lock' if submission of the revision is attempted 21 or more days after the deadline. If you do not think you will be able to meet this deadline please contact the editorial office immediately.

on behalf of Professor Joshua Ross (Associate Editor) and Marta Kwiatkowska (Subject Editor)
openscience@royalsociety.org

Reviewer comments to Author:

Reviewer: 1
Comments to the Author(s)
See attached pdf file ("aspinall_rep.pdf").

Reviewer: 2
Comments to the Author(s)
Comments attached ("Review RSOS.pdf").

===PREPARING YOUR MANUSCRIPT===

Your revised paper should include the changes requested by the referees and Editors of your manuscript. You should provide two versions of this manuscript and both versions must be provided in an editable format:
one version identifying all the changes that have been made (for instance, in coloured highlight, in bold text, or tracked changes);
a 'clean' version of the new manuscript that incorporates the changes made, but does not highlight them. This version will be used for typesetting if your manuscript is accepted.

===PREPARING YOUR REVISION IN SCHOLARONE===

<https://royalsociety.org/journals/authors/author-guidelines/#supplementary-material> to include a suitable title and informative caption. An example of appropriate titling and captioning may be found at https://figshare.com/articles/Table_S2_from_Is_there_a_trade-off_between_peak_performance_and_performance_breadth_across_temperatures_for_aerobic_scops_in_teleost_fishes_/3843624.

Author's Response to Decision Letter for (RSOS-202218.R0)

See Appendices C & D.

Decision letter (RSOS-202218.R1)

Dear Dr Aspinall

On behalf of the Editors, we are pleased to inform you that your Manuscript RSOS-202218.R1 "Pupils returning to Primary Schools in England during 2020: rapid estimations of punctual COVID-19 infection rates" has been accepted for publication in Royal Society Open Science subject to minor revision in accordance with the referees' reports. Please find the referees' comments along with any feedback from the Editors below my signature.

Please submit your revised manuscript and required files (see below) no later than 7 days from today's (ie 24-Aug-2021) date. Note: the ScholarOne system will 'lock' if submission of the revision is attempted 7 or more days after the deadline. If you do not think you will be able to meet this deadline please contact the editorial office immediately.

on behalf of Professor Joshua Ross (Associate Editor) and Marta Kwiatkowska (Subject Editor)
openscience@royalsociety.org

Associate Editor Comments to Author (Professor Joshua Ross):
Associate Editor

Comments to the Author:

Thanks for your responses to the Reviewers' comments. I support your view that the majority of Reviewer 2's comments need not be addressed explicitly, due to a mis-match in the timing of data availability and manuscript production, and the purpose of the framework in regard to rapid analysis to support decision making.

However, I would like you to consider adding a sensitivity analysis to the paper, in which you assess both the incidence-prevalance and steady-state assumptions. It is my understanding that knowing the impact of these assumptions would be useful to guide the potential use of your framework in future outbreaks; and that your decision to not do this was motivated by Reviewer 2 also wanting you to shorten the length of the paper.

===PREPARING YOUR MANUSCRIPT===

===PREPARING YOUR REVISION IN SCHOLARONE===

<https://royalsociety.org/journals/authors/author-guidelines/#data>. You should ensure that

you cite the dataset in your reference list. If you have deposited data etc in the Dryad repository, please only include the 'For publication' link at this stage. You should remove the 'For review' link.

Author's Response to Decision Letter for (RSOS-202218.R1)

See Appendix E.

Decision letter (RSOS-202218.R2)

Dear Dr Aspinall,

I am pleased to inform you that your manuscript entitled "Pupils returning to Primary Schools in England during 2020: rapid estimations of punctual COVID-19 infection rates" is now accepted for publication in Royal Society Open Science.

COVID-19 rapid publication process:

We are taking steps to expedite the publication of research relevant to the pandemic. If you wish, you can opt to have your paper published as soon as it is ready, rather than waiting for it to be published the scheduled Wednesday.

This means your paper will not be included in the weekly media round-up which the Society sends to journalists ahead of publication. However, it will still appear in the COVID-19 Publishing Collection which journalists will be directed to each week (<https://royalsocietypublishing.org/topic/special-collections/novel-coronavirus-outbreak>).

If you wish to have your paper considered for immediate publication, or to discuss further, please notify openscience_proofs@royalsociety.org and press@royalsociety.org when you respond to this email.

on behalf of Professor Joshua Ross (Associate Editor) and Marta Kwiatkowska (Subject Editor)
openscience@royalsociety.org

Appendix A

Quantifying risk levels in Primary Schools in England from COVID-19 infection hazard by Aspinall *et al.*

Referee's comments for authors

Note: Line numbers are those on the inside numbering, except where these are absent.

- (1). In many places you use the term “odds” as if it were a synonym for “probability”. It is not. Please do a global replace to correct.
- (2). p.6, line 194. Whether or not this “companion paper” is published and accessible to readers, it would be good to have more information (perhaps in the supplementary material) about the conduct and output of this elicitation process.
- (3). p.7, Figure 1. I think the node “Children_prevalence” should not have handles.
- (4). p.7, line 236. What is the interpretation of this probability density function?
- (5). p.7, line 242 *ff.* In modelling Scenarios IIIa–c, I would realistically expect you to add further uncertainty about assumed prevalence. At least comment on this.
- (6). p.8, line 255. The assumption of a steady state seems very dubious.
- (7). p.8, line 272 *ff.* Say some more about how this distribution was constructed. Was it entirely data-driven, or was there some expert opinion input? If the latter, say more about the elicitation process, and in any case discuss the strengths and weaknesses of the overall procedure.
- (8). p.9, line 294. Again, more information is needed about the procedure(s) used for quantifying parameter uncertainty (perhaps give details in the supplementary material.)

- (9). p.9, line 308. This point could do with explanation and expansion.
- (10). p.11, Table 2b. Column heading “Number” should be “Mean”.
- (11). p.12, line 374 *ff.* “over-stated” is an odd usage. At the time it was announced, the uncertainty should have been correctly stated. If as a result of further information (or speculation) opinions have changed, that is a different matter. In any case I find the conditioning on being above or below the prior median an odd thing to do, since it does not seem to relate clearly to new information that might become available. I find it difficult to see how this kind of analysis could be useful for setting policy. Please supply further justification, or at least admit that this is a bit of a “kludge”.
- (12). p.12, line 389. “converse” → “reverse”.
- (13). p.13, line 400. “counterfactual” → “hypothetical” (nothing counter to known facts is involved in considering possible future scenarios).
- (14). p/13, line 411. “there” → “their”.
- (15). p.15, line 470. “do” → “so”.
- (16). p.15, line 497. “half-log” needs explanation.
- (17). p.15, line 499. Did you just scale the overall distribution, without allowing for realistic further uncertainty? In any case there is no Table 1 in my copy (and if there is to be such a table, it should probably be numbered differently).
- (18). p.16, Table 5a. It would be better to avoid the tiny numbers, and give (say) rates per 100,000. This might overcome the anomaly of giving 0 as a 5th percentile.
- (19). p.21, line 680 *ff.* More detail about the provenance (and realism) of these distributions, please.
- (20). p.21, line 693. Delete “quantiles”.
- (21). p.25, line 22. “there” → “the”.
- (22). p.25, line 42. “were” → “was”.

Appendix B

To the Authors:

The authors use an BBN framework to project the number of primary schools in England with 1 or more covid-19 cases during 5 scenarios, addressing the restrictions imposed in the UK during June, July and September. They address how the number of pupils, number of teaching staff and other staff in England changes based upon the community level of prevalence, finding that adult prevalence in the community has a stronger effect on the projected infection numbers than the adult to child prevalence ratio. This paper is a demonstration of the UNINET software for epidemiological usage. We do feel that there is potential for a really nice paper here, that will be of interest to researchers studying covid infection in schools and researchers studying BBNs, however we find there to be a couple of substantial issues with the paper that prevent me from recommending it for publication at this point.

Broadly, these can be grouped into four areas: i) issues with the structure of the paper (in particular streamlining the presentation of the methods and results) ii) ambiguity in the methods iii) no referencing to other studies presented in the literature and iv) the inclusion of certain conclusions that we are not convinced are backed up by the results shown, and we are concerned are in fact erroneous.

Major

Main conclusions, comparing projections to data

This paper would benefit from restructuring Section 6, Results, by integrating the “Comparison with observation” subsection. Instead the authors should refer and discuss how their results compare to observed data throughout the text.

A substantial issue with the results of this paper, is how they compare to the observed spread of COVID-19 in schools. The authors predicted that 1925 schools would be infected in July 2020, and compared this to the PHE data (116 positive covid incidents from mid May to end of June). The authors suggest that the discrepancy in their results and the observation, is due to asymptomatic cases, poor reporting and reporting errors (line 634), while all these limitations in the PHE data could be present, the authors have not reflected on any limitations of their method.

Instead, the authors compare their results on the mean number of infected schools in England in September to the DfE report on the number of students absent from school due to self-isolating from a covid-19 contact within school (line 720). The authors find an agreement between their model output (3,087 infected schools) and the DfE data (3,000 children isolating due to an infected contact), however this is fundamentally incorrect. We are under the impression that their model doesn't project results of self-isolation within schools, and instead their model looks at the number of infected persons within a school. It is not clear what the link on line 709 is which allows the authors to equate prevalence to self isolation. This is comparing two different data types, and indeed it should be expected that the number of children self isolating is far higher than the number of infected individuals within a school. Especially since the control measures implemented in the UK resulted in entire year groups being sent home to self-isolated when there was a single positive case identified. Additionally, the authors do not also consider the DfE data on the number of staff isolating due to a contact with an infectious covid-19 case in school, which is also available.

It is unclear why the authors didn't use data from DfE (same source) on the number of pupils, teachers and other staff who are absent with a confirmed or suspected covid-19 infection. This data could be used to approximate the number of schools with an infected person in the UK, and would be a better comparison to their model output.

Ambiguity in the methods

It is unclear if the analysis by the authors is computed over the whole month of June, July and September; over the whole term or for a single date. Line 655 (page 20) suggests it is the latter and only three single dates of the ONS prevalence sample survey are considered, namely Monday 1st June 2020, Friday 5th June 2020 and Thursday 10th September 2020. This is a major issue, as these dates aren't representative of the whole time frame, especially given the authors conclude that prevalence in the whole of September was 2.5 times higher than the prevalence in June. This is repeated throughout the text, when referring to "june-levels", "june prevalence", or "september-levels" in Section 6, Results; as such it is unclear if Table 2a shows the number of infected schools at the start of the month, totalled over the whole month or the whole term. This needs to be made clear in the text and in the caption of all tables and figures.

Further detail is required to explain why data from June 5th was representative, line 322 suggests that it is the maximal value rather than a representative one. While not perfect, have the authors considered taking the average prevalence in June which would be more representative than considering a single day? Indeed, reporting errors are not consistent throughout the week in the UK, with fewer infections being recorded over the weekend (e.g. Friday-Monday), and a much higher reporting rate on Tuesdays. Taking the average in June, could minimise these errors. Additionally, the authors need to support their decision of selecting September 10th; in England, all students were welcomed to return to schools from September, but many schools chose to have a gradual return, particularly for nursery and Year 1. This gradual return is not unique to 2020 and occurs every year in September over the first 2-3 weeks (i.e. including 10th September). For scenario III, if the authors did not consider the delayed reopening, they are likely overestimating the number of infected persons in schools.

On a related note, on line 321 the authors refer to the temporal trends being noisy in the ONS prevalence data, as so do not present a temporal analysis. However, if your model can be evaluated daily to provide temporal trends, such as the number of infected schools over time, even if it is noisy, we think this would be very beneficial to the reader. This could be presented as a time series figure, demonstrating how infection in schools projected by the BBN model changes over time, depending on the prevalence inputted. Additionally, this calculation can then replace the questionable approximation on line 622, which finds the total number of infected schools in June and July by extrapolating the results evaluated on 1st June, using the daily rate of new infected schools (rate =55, line 629, not referenced in text!) multiplied by the average infectious period. This approximation not only uses a rate which is not explained in the text, but also it does not take into account that infected individuals within a school would be isolated, and so the infectious duration of a school is not the same as the infectious duration of an individual. Instead, the authors could run their model for each day in June and take the summation, as well as presenting a time series visualisation of this data.

On line 252, the authors describe using incidence as a proxy for prevalence. This is not a recommended approximation, please reference in the text papers which present the limitations of this method. In particular, the authors state “to first order, prevalence can be related to incidence since, in a steady state, numbers of recovered individuals who are no longer infectious will balance numbers of new infections” on line 256, and we are under the impression that this approximation is only true where the disease has life long immunity. Life long immunity is not thought to be true for COVID-19 and the length of the waning immunity is disputed. Additionally, the author's statement is only reasonable at steady state, but we don't think it is reasonable to assume that the UK was at a steady state in June 2020. If the authors are going to take this approximation, we would like to see some sensitivity analysis performed to determine the impact of this on your results. Additionally, even assuming this proxy, it is still unclear how incidence was used to calculate prevalence. Details of this transformation should be commented on in the main text (1 sentence max), in particular was the average infectious period used to evaluate prevalence from incidence? Was the delay in reporting new cases, or the delay in presentation of symptoms considered in this approximation?

In Appendix A1: Incidence and Prevalence, the authors use a factor of 6x to reconcile incidence and prevalence, however this value has not been referenced. While the authors address that there are uncertainties in method, they have not illustrated the potential impact of these uncertainties. We think you need to perform a sensitivity analysis for this factor.

Condensing the script, removing exploratory analysis, readability

As currently written, the manuscript is very repetitive. This is one of the main factors contributing to its length. We recommend that the manuscript be edited down to a more reasonable length, which should be achievable if the sections repeating what has already been said are removed. For example, using incidence data on UTLA as a proxy for prevalence is explored several times in the text. Due to its length, it is very difficult to find the main findings. Instead, the manuscript reads as a large collection of results, without much thought being given to which ones are most important.

As mentioned above, a lot of the initial analysis/ exploration is just a subset/ specific version of later results. These earlier results, while useful in your exploration, are not required in the manuscript, and just add to the already long manuscript. Large sections of the manuscript could also be moved to the appendix, for example, section 6.3 and table 3. Similarly, why does the paper initially focus on using the mean school size, when you point out that schools vary significantly (page 18, line 581). This work could be significantly condensed by removing all exploratory analysis using the simplified assumption on the mean school size, and only presenting the analysis with school size drawn from the national distribution of school sizes.

Additionally, we are concerned about the number of times the appendix is cross referenced. Many of the important assumptions, parameters and methodology of your work are in the appendix, making it a difficult paper to read. In particular, parameters (e.g. infection periods) should be listed in a table in the main text, as well as information of schools such as on page 7, line 280 average pupil headcount. If you have to repeatedly cite the supplementary

material, then it should be contained in the main text. The following sections should be reformatted (condensed) and moved to the main part of the paper:

- Include a table of data sources
- Table of epidemiological parameters and sources
- Couple of sentences of how prevalence data was obtained from incidence data (what parameters you used)
- Move assumptions into the main text

The following sections should be removed from the main text and placed in the appendix:

- Results: all results using a fixed school size for all schools. Re-create Figure 2 for school sizes drawn from a distribution.
- What if? Sensitivity tests: move this section and all tables to appendix. Replace with a single sentence referencing this work with its main finding (lines 414-416) that adult prevalence has a stronger effect on projected infection numbers.

Rather than considering the comparisons to true recorded data at the end, include the prevalence in September being 2.5 times greater than the June prevalence from the beginning and replace the “scenario 4x” to “scenario 2.5x”. There’s no need to consider the “4x” or “1/4x” scenarios when we know the ratio was actually 2.5.

In your spatial analysis, does each half-log prevalence bin correspond to a specific region in the UK? This could be important in terms of the number of students eligible to attend school in June 2020 varying spatially. We also have the following concerns with your spatial analysis:

- Page 15, line 497, not clear what six half-log incidence bins with sampling uncertainties are. My understanding is that you are dividing England into 6 groups based upon the prevalence in UTLA, and table 5a shows the number of UTLAs in England which has a very-high prevalence level in bin 6 and UTLAs with low prevalence in bin 1? We would name each bin by categories which are more intuitive such as “very-high prevalence”, “high”, “medium”, “medium-low”, “low” and “very-low”.
- This isn’t really a spatial analysis. You are considering 6 independent regions with different community prevalences, rather than a number of potentially interacting populations (e.g. a low prevalence region surrounded by high prevalence regions could behave differently to a low-prevalence region surrounded by low-incidence regions). As far as we can tell, you have essentially just performed a sensitivity analysis on the community prevalence.
- We don’t think Table 5a is needed in the text (move to appendix) instead just include in the main text the population weighting of each category.
- For the spatial study, Page 15, line 500 - have you calculated the number of positive tests or the proportion of positive tests (i.e. positive/(positive+negative)). Given the heterogeneous nature of testing in England, the analysis needs to be done on the proportion of positive tests.

Referencing literature and sources

The manuscript currently looks very little into the existing literature on covid-19 in schools. The work would greatly benefit from a review of the current literature. As such, a section

describing how this work compares to more traditional epidemiological models, and how the authors' work fits into the literature would improve the manuscript. As it stands, almost a quarter of the references are from the authors' works - is this representative of the covid-19 literature?

A large number of parameters are introduced, without any citations for them. If the values are not taken from a particular source, a justification (and ideally a sensitivity analysis) should be provided for them. For example, line 203, you include a +/- 10% uncertainty on the estimates. Why 10%? Is this a standard value to take? It's not clear that this will be sufficient to cover the part time staff and teaching staff on rota. There are many other similar occurrences in your manuscript. As stated elsewhere in this review, this would be most convenient listed in a table along with all the other parameters you've used, rather than scattered across the main text and appendix.

There are a number of other places where citations should be included for the claims you have made, including but not limited to the following:

- Page 3 of 27, line 77, cite to references
- Page 4 of 27, line 103, cite other studies on this topic
- Page 6 of 27, line 193, Primary and secondary, and establishment type (state, private etc) is available on the Gov website.
- Page 6 of 27, line 211, reference where you got the susceptibility ratio
- Page 6 of 27, line 184, how do you calculate your contact matrix. Provide reference to other contact matrix surveys. What is your contact matrix?
- Page 7 of 27, line 240, reference the typical absenteeism value
- Page 9, line 318, the value of 65% staff returning to school in June, cite this. Or is this an approximation? If latter, what was actually observed?
- Page 15, line 492, citation needed
- Page 15, line 495, citation for the regional prevalence in high-risk regions being 350 times worse

Figures

- Figure 1: Why is children_returning_count in green? What is the reasoning behind the stdNorm_distribution? The quality of this figure is very blurry.
- Results: figure 2, remove the border of this figure and include the mean/std as a legend in the plot. Provide y labels and x labels. Improve image quality (very blurry). This figure shows the distribution for Scenario IIIb, we think your results section would improve greatly if you included this figure for each of the scenarios.

Other

We have included below a number of other suggestions that we didn't think neatly fell into one of the above categories.

Could you clarify how you have produced table 2b in your results section? Is this the number of primary school children you would expect nationally or per school? We assume the former, if the former (over the whole of England) changes this table from the number of pupils to the proportion of pupils, teachers and ancillary staff. It would be interesting to know

the proportion of teachers rather than raw numbers as this can be compared to the national prevalence. Again, change tables 3 and 4 to use proportions.

Could you clarify what table 5b shows? Is the spatial prevalence in this table showing the number of primary schools in England after using a different prevalence for each binned region? Table 6 is far more interesting, presenting the proportion of schools in each binned category which had 1 or more infected case, not just the total in England. Table 5c is comparing a non-spatial prevalence to a spatial-prevalence method, without providing any insight on the spatial heterogeneity of covid cases within schools. As such, this should be moved to the appendix.

We are also concerned that in Table 6, you do not consider the total number of pupils, teachers, staff in each of the binned sections and assume that this is constant for all binned sections? We would like to see this carried out for a non-average sized school. Likewise with Table 5c, rather than aggregating your spatial results over the whole of England, it would be more interesting to present the number of infected persons for each binned region. Change table 5c to proportion of pupils in each binned region which are infected, and move to the appendix.

On page 18, line 585, you say there is no change in the overall number of infected persons when taking the mean school sizes vs school sizes drawn from the national distribution however there is no table to back up this statement. We think it would be best to remove all work using the mean school size, and just focus on national distribution.

On page 19, could you clarify how the mean probabilities were found? Remove all lists of $N=0$, Prob = ... into a single table, so there is a column in the table for each scenario considered.

Minor

- There is no Table 1? Document starts with Table 2a, although the generic scaling factor is in table 1 (page 15, line 499)
- Page 3 of 27, line 48 and 49 “The devolved administrations... re-open their schools.” Don’t think this is necessary
- Page 5 of 27, line 140, assume the audience does not know what an “acyclic graph” is.
- Page 5 of 27, line 143 why is the BBN method only “quite general”
- Page 6, line 213 this reference to background about BBN should be in section 3. Methodology where BBN is introduced
- Page 9, line 306, add to this paragraph what the figure shows, i.e. the distribution of the number of primary schools with at least one infection.
- Page 12 line 363, sensitivity analysis is common amongst many epidemiological models and isn’t unique to BBN. Rephrase/remove lines 363 to 365.
- Move the results on the probability of having two or more cases, or three or more cases to earlier. The rest of section 6.3 can be moved to the appendix.
- As a related aside, in figure 1 a fixed input value for prevalence is taken, I wonder whether the confidence interval of prevalence should be considered here.

Appendix C

University of
BRISTOL

as from
Department of Earth Sciences
University of Bristol
Queen's Road
Bristol BS8 1RJ

The Editor,
Royal Society Open Science.
11 August 2021
By email

Dear Editor

Manuscript ID RSOS-202218: Pupils returning to Primary Schools in England during 2020: rapid estimations of punctual COVID-19 infection rates *

Authors: Aspinall, W.P., Sparks R.S.J., Woodhouse, M.J., Cooke, R.M., Scarrow, J.H. and the CoMMinS Project

We are very grateful to the reviewers for their time and conscientious efforts in reviewing our manuscript and to you for your assistance and patience. We provide detailed responses to the reviewers' comments in a separate document.

*Reflecting on both reviewers' comments overall, we think this modified title may help clarify the essence of our contribution. We hope this change is acceptable?

We are happy to accommodate almost every one of Reviewer #1's suggestions and recommendations which, we are pleased to acknowledge, help improve the manuscript.

Reviewer #2 provided extensive comments and suggestions, for which we are also grateful. However, from these comments we surmise that Reviewer #2 has in mind that our study would be better developed into a *post hoc* epidemiological research paper, whereas we were seeking to report work on an immediate risk assessment analysis, performed in real-time during early stages of the COVID-19 crisis. At the time, many of the data sources Reviewer #2 mentions did not yet exist.

The coronavirus pandemic outbreak presented many novel challenges in a public health context and we endeavoured to leverage our experience - as volcanologists, probabilists and stochastic modellers - of assessing dangerous societal threats under circumstances of extreme uncertainty. A closely linked companion paper on other aspects of the same issue has been published in RSOS¹. Jointly, these two contributions reflect the risk assessment challenge we confronted under the Royal Society RAMP initiative. As far as we know, these calculations of primary schools' infection risks were the only punctual forecast estimates made ahead of returns to school in June and September 2020.

Because there is this dichotomy between what Reviewer #2 recommends and what we did for our risk assessment, we find it very difficult to agree to all the many additions and substantive changes suggested by Reviewer #2 – we are happy to implement several, but regret we cannot accede to all. And, we may add, that while Reviewer #2 criticises the

¹ Sparks RSJ et al., 2021. "A novel approach for evaluating contact patterns and risk mitigation strategies for COVID-19 in English primary schools with application of structured expert judgement". R. Soc. open sci.8201566201566

length of our manuscript, they also suggest multiple items of further work, entailing extra analyses, a comprehensive literature review and additional topics for reportage and discussion.

Kindly note that we have added the Bristol University CoMMinS Project to the list of authors; in preparing our responses, we benefitted from discussions with a number of specialist colleagues in that project.

We look forward to hearing your editorial decision in due course.

Yours sincerely,

Prof. Willy Aspinall.
Emeritus Professor, University of Bristol
Email: Willy.Aspinall@Bristol.ac.uk

Cc Sparks; Woodhouse; Cooke; Scarrow; Relton (for CoMMinS)

Appendix D

Manuscript ID RSOS-202218: Pupils returning to Primary Schools in England during 2020: rapid estimations of punctual COVID-19 infection rates *

Authors' response to RSOS Editor

Authors: Aspinall, W.P., Sparks R.S.J., Woodhouse, M.J., Cooke, R.M., Scarrow, J.H. and the CoMMInS Project

We are very grateful to the reviewers for their time and conscientious efforts in reviewing our manuscript and to you for your assistance and patience. We provide detailed responses to the reviewers' comments in a separate document.

*Reflecting on both reviewers' comments overall, we think this modified title may help clarify the essence of our contribution. We hope this change is acceptable?

We are happy to accommodate almost every one of Reviewer #1's suggestions and recommendations which, we are pleased to acknowledge, help improve the manuscript.

Reviewer #2 provided extensive comments and suggestions, for which we are also grateful. However, from these comments we surmise that Reviewer #2 has in mind that our study would be better developed into a post hoc epidemiological research paper, whereas we were seeking to report work on an immediate risk assessment analysis, performed in real-time during early stages of the COVID-19 crisis. At the time, many of the data sources Reviewer #2 mentions did not yet exist.

The coronavirus pandemic outbreak presented many novel challenges in a public health context and we endeavoured to leverage our experience - as volcanologists, probabilists and stochastic modellers - of assessing dangerous societal threats under circumstances of extreme uncertainty. A closely linked companion paper on other aspects of the same issue has been published in RSOS . Jointly, these two contributions reflect the risk assessment challenge we confronted under the Royal Society RAMP initiative. As far as we know, these calculations of primary schools' infection risks were the only punctual forecast estimates made ahead of returns to school in June and September 2020.

Because there is this dichotomy between what Reviewer #2 recommends and what we did for our risk assessment, we find it very difficult to agree to all the many additions and substantive changes suggested by Reviewer #2 – we are happy to implement several, but regret we cannot accede to all. And, we may add, that while Reviewer #2 criticises the length of our manuscript, they also suggest multiple items of further work, entailing extra analyses, a comprehensive literature review and additional topics for reportage and discussion.

Kindly note that we have added the Bristol University CoMMInS Project to the list of authors; in preparing our responses, we benefitted from discussions with a number of specialist colleagues in that project.

We look forward to hearing your editorial decision in due course.

Willy Aspinall, for authors

Authors' responses (in blue type) to Reviewer comments to Author

Reviewer: 1

Comments to the Author(s)

Quantifying risk levels in Primary Schools in England from COVID-19 infection hazard
by Aspinall et al.

Referee's comments for authors

Note: Line numbers are those on the inside numbering, except where these are absent.

(1). In many places you use the term “odds” as if it were a synonym for “probability”. It is not. Please do a global replace to correct.

Changes made to six instances in text.

(2). p.6, line 194. Whether or not this “companion paper” is published and accessible to readers, it would be good to have more information (perhaps in the supplementary material) about the conduct and output of this elicitation process.

In Supplementary Information Appendix A1, we have added a summary of the now-published elicitation methodology used by Sparks et al. (2021) (ref [1]) and those key findings which here inform our numerical modelling.

The main text now reads:

Instead, we drew on information from an elicitation of teachers in thirty-six Primary Schools in the Royal Society Schools network, described in a companion study [1]; key aspects of that elicitation study are reported in Appendix A1.

(3). p.7, Figure 1. I think the node “Children prevalence” should not have handles.

Figure 1 is correct. In the absence (at the time) of established data on prevalence in children, the node <Children_prevalence> is a calculational node, a joint function of known data and uncertainties for <Adult_prevalence>, multiplied by an uncertainty distribution for <Adult_child_prevalence_ratio> centered on the, then current, generic belief about relative infection rates for children in relation to infected adults.

(4). p.7, line 236. What is the interpretation of this probability density function?

We explain this now with amended text in the last sentence:

Scenario I considered the situation on 1st June 2020, the first day of school return for restricted sets of Year group children, as stipulated by the UK Government. Numbers of persons attending school were obtained from ONS data, who also published a probability density function for adult prevalence on 5th June 2020 (mean value: 1 in 1700); the statistical parameters of the ONS PDF were adopted in the BBN model <Adult_prevalence> node to characterise prevalence uncertainty for Scenario I.

(5). p.7, line 242 ff. In modelling Scenarios IIIa–c, I would realistically expect you to add further uncertainty about assumed prevalence. At least comment on this.

In July 2020, there was substantial forward uncertainty as to what the community prevalence would be when schools returned fully in September 2020. Rather than inflating the June 2020 ONS PDF uncertainty by an arbitrary amount, we decided to offer 2 alternative scenarios for full school returns in September 2020 by multiplying and dividing the June 2020 prevalence ratios by 4. Scenarios IIIa–c were therefore chosen to provide three indicative, but plausible, future states of community prevalence for school returns in the

(then) forthcoming new academic year. This approach also reflects the burden on decision-makers when interpreting uncertain forecasts: adding stochastic uncertainty on prevalence values is certainly feasible within the model but making sense of the results becomes more difficult as these uncertainties are convolved with other stochastic components of the model. Fixed scenario-based forecasts allow direct interpretation and, while they are unlikely to capture the exact circumstances, support decision-makers in planning for plausible future outcomes.

The text now reads:

Looking forward to the full re-opening of schools in September, with all Year groups, Scenario IIIa assumed the return of all Primary School children, adjusted for typical absentee rates. Given substantial forward uncertainty in July 2020 concerning the projected community adult prevalence level in September 2020, rather than simply inflate the ONS PDF by an arbitrary amount we chose to model the potential future prevalence by two additional marker sub-scenarios: one (Scenario IIIb), in which the adult prevalence in the community in September was assumed 4x greater than that given by ONS from their 5th June 2020 survey, and another for a drop in September to $\frac{1}{4}$ of the June prevalence (Scenario IIIc). We considered that this pair of sub-scenarios represented two contrasting but plausible prospective levels of adult prevalence in September 2020, with associated uncertainties. The corresponding total numbers of persons in schools for each Scenario are given in Supplementary Information Appendix A1 Table A1.2.

(6). p.8, line 255. The assumption of a steady state seems very dubious.

This was another assumption that we felt was justified in the circumstances as it allowed our calculations to be tractable on an urgent timescale. Going further would require more complex calculations, likely requiring other epidemiological models to be implemented and therefore propagating many more uncertainties into the calculations without necessarily adding any substantial information gain.

Text now reads:

To first order, prevalence can be related to incidence since, in a steady state, numbers of recovered individuals who are no longer infectious will balance numbers of new infections (while this is a questionable assumption, it allowed immediate model calculations to be tractable). Thus, local or regional prevalence can be scaled from local or regional incidence data: the basis for this conversion is outlined in Supplementary Information Appendix A1.

(7). p.8, line 272 ff . Say some more about how this distribution was constructed. Was it entirely data-driven, or was there some expert opinion input? If the latter, say more about the elicitation process, and in any case discuss the strengths and weaknesses of the overall procedure.

The uncertainty distribution we chose was our synthesis of the diverse findings of a number of studies; it did not involve formal elicitation or meta-analysis. Because our study was, in effect, an immediate risk assessment, we adopted the philosophy that it is a defensible approach to adopt such uncertainty characterizations for some factors, given all distributions are enumerated and recorded in our model (i.e. in Appendix A3). Alternative representation(s) can be constructed, but with costs of time and effort and perhaps little information gain. However, one strength of our approach is that such analyses can be performed straight-forwardly using the facilities of the UNINET software.

We have modified the text to explain this more clearly, as follows:

A key issue was that, at the time we performed our analysis, there was mounting evidence that children might be less susceptible to COVID-19 infection than adults. The comprehensive modelling analysis of global data by [7] led us to adopt a probability distribution function for the ratio of adult to children's susceptibility with a median of 2.3 and 90% credible interval between 1.3 and 3.8. This range reflected our synthesis of results from other studies in order to characterize, in a single distribution for modelling purposes, the susceptibility ratio between adults and children (e.g. [8–11]). For an urgent risk assessment, we consider this a defensible approach for characterizing the uncertainties associated with some factors; all the distributions used in our model are recorded in Appendix A3).

(8). p.9, line 294. Again, more information is needed about the procedure(s) used for quantifying parameter uncertainty (perhaps give details in the supplementary material.) Information about our procedures for quantifying parameter uncertainties is in the various sections where different variables and parameters are introduced and discussed. All distribution enumerations are recorded in Appendix A3, and we have added a phrase in text at about line 300 to point the reader to these details:

As a basis for enumerating quantitative risk assessments and informing policy decisions, it is essential that parameter uncertainties are properly accommodated in the model and that their influence on outputs are fully articulated; numerical details of all distributions used in the model are recorded in Appendix A3.

(9). p.9, line 308. This point could do with explanation and expansion.

In concert with a related comment by Reviewer #2, we have modified the text, as follows: An example model output distribution is shown in Figure 2, showing the distribution of number of schools with at least one infection under Scenario IIIb (above). This plot illustrates one typical BBN result with typical uncertainty spread and some skewness in distribution shape; such long-tailed behavior, as evinced here, could have implications for policy decision-making. This is pertinent because a societal risk assessment that is expressed solely as a central tendency estimate (e.g. as a mean or median, alone) may not be adequate or appropriate for high-consequence risk-informed decision making. Therefore, it is crucial to express such assessment results with a suitable set of confidence levels or exceedance probabilities, such that the decision maker can have a full probabilistic context for appraising the conservatism and tolerability, or otherwise, of the threat.

(10). p.11, Table 2b. Column heading “Number” should be “Mean”.

Tables 2a & 2b both modified to read “Mean number”.

(11). p.12, line 374 ff. “over-stated” is an odd usage. At the time it was announced, the uncertainty should have been correctly stated. If as a result of further information (or speculation) opinions have changed, that is a different matter. In any case I find the conditioning on being above or below the prior median an odd thing to do, since it does not seem to relate clearly to new information that might become available. I find it difficult to see how this kind of analysis could be useful for setting policy. Please supply further justification, or at least admit that this is a bit of a “kludge”.

In relation to “over-stated” uncertainty, we had in mind the second situation that the reviewer mentions: i.e. the possibility of a subsequent change in parameter uncertainty characterisation due to later or better information becoming available (a not unusual circumstance in our

experience with urgent risk assessments). The para in question may be not well-expressed, and therefore we make some changes:

There can be several reasons for exploring what-if? scenario impacts. For instance, there may be surrogate evidence, say from another country or region, suggesting that a particular parameter likely falls within a certain limited range, when local data are inadequate for constraining that range. The potential implications for our model results -- of learning, later, that the uncertainty on the relevant parameter was originally over-stated -- can be appraised by conditionalizing on the surrogate range. The UNINET software is ideally suited to this sort of diagnostic what-if? analysis.

The remainder of the reviewer's comment we find challenging, and not consonant with our own (quite extensive) experience of communicating risk assessment findings to decision makers: the latter not infrequently ask "what would be the impact on your results if such-and-such a parameter is higher (or lower) than your central value?". In situations where new or revised data cannot be waited for, especially those involving predictions of uncertainty, it can be expedient to run some quick, alternative "what if?" scenario tests for decision makers to consider; any decisions based on such tests are the responsibility of the decision maker – the analyst's role is to provide as much information in the wider context as is advantageous to the decision maker.

Because this is fundamental aspect of risk assessment communication, therefore we suggest adding the following para to the text:

With societal risk assessments, decision makers quite often ask "what would be the impact on your results if such-and-such a parameter is higher (or lower) than your central value?". In situations such as this, where new or revised data cannot be waited for, it can be expedient to run some quick, alternative "what if?" scenario tests for decision makers to consider, especially for parameters or variables that rely on predictions of uncertainty. The risk analyst's role is to provide as much, and only as much, information in the wider context as is advantageous to the decision maker. This said, policy decisions based on such tests remain the responsibility of the decision maker -- the analyst should provide neutral and balanced alternative what-if? scenarios.

We think it might be helpful also to change the section heading to:

6.2 'What if? scenario tests

(12). p.12, line 389. "converse" → "reverse".

With a little reluctance, we change to "reverse".

(13). p.13, line 400. "counterfactual" → "hypothetical" (nothing counter to known facts is involved in considering possible future scenarios).

While the meaning here is debatable, we make the recommended change.

(14). p/13, line 411. "there" → "their".

Thank you.

(15). p.15, line 470. "do" → "so".

Thank you.

(16). p.15, line 497. "half-log" needs explanation.

Text changed to "six half-log (**10^{0.5}-fold**) incidence bins".

(17). p.15, line 499. Did you just scale the overall distribution, without allowing for realistic further uncertainty? In any case there is no Table 1 in my copy (and if there is to be such a table, it should probably be numbered differently).

Well spotted! Table 1 is Table A1.3 in Supplementary Information Appendix A1. Text changed accordingly.

When the study was being implemented, the overall incidence distribution in Appendix A1 was converted to prevalence by uniform scaling; further uncertainties are not included (but would be desirable). The modified text now reads:

As a simple illustration, we explored how this variation might affect our projected results. In the BBN, six half-log (i.e. $10^{0.5}$ -fold) incidence bins with sampling uncertainties were introduced, with separate contiguous weekly incidence rates, and these were each converted to equivalent prevalence distributions using a generic, uniform scaling factor without allowing for further uncertainties (see Table A1.3 in Appendix A1); prevalence values are the inverse of the probability of there being an infected person in a population. For each bin, we calculated from ONS data the total number of persons that tested positive in each bin to estimate a sampling error.

(18). p.16, Table 5a. It would be better to avoid the tiny numbers, and give (say) rates per 100,000. This might overcome the anomaly of giving 0 as a 5th percentile.

Table and caption modified, as suggested.

(19). p.21, line 680 ff. More detail about the provenance (and realism) of these distributions, please.

Please see our response to comment (5), above. In pursuit of rapid, indicative model results for urgent decision support, we were obliged to make pragmatic choices about characterizing some BBN node distributions where data was sparse, unreliable or non-existent. We think the existing text and discussion adequately reflects this context.

(20). p.21, line 693. Delete “quantiles”.

Deleted.

(21). p.25, line 22. “there” → “the”.

Thank you.

(22). p.25, line 42. “were” → “was”.

Thank you.

Reviewer: 2

Comments to the Author(s)

To the Authors:

The authors use an BBN framework to project the number of primary schools in England with 1 or more covid-19 cases during 5 scenarios, addressing the restrictions imposed in the UK during June, July and September. They address how the number of pupils, number of teaching staff and other staff in England changes based upon the community level of prevalence, finding that adult prevalence in the community has a stronger effect on the projected infection numbers than the adult to child prevalence ratio. This paper is a demonstration of the UNINET software for epidemiological usage. We do feel that there is potential for a really nice paper here, that will be of interest to researchers studying covid infection in schools and researchers studying BBNs, however we find there to be a couple of substantial issues with the paper that prevent me from recommending it for publication at this point.

Broadly, these can be grouped into four areas: i) issues with the structure of the paper (in particular streamlining the presentation of the methods and results) ii) ambiguity in the methods iii) no referencing to other studies presented in the literature and iv) the inclusion of certain conclusions that we are not convinced are backed up by the results shown, and we are concerned are in fact erroneous.

Response (23) We thank the reviewer for their positive, constructive and critical comments. We respond to these generalised criticisms in detail, below.

Major

Main conclusions, comparing projections to data

This paper would benefit from restructuring Section 6, Results, by integrating the “Comparison with observation” subsection. Instead the authors should refer and discuss how their results compare to observed data throughout the text.

A substantial issue with the results of this paper, is how they compare to the observed spread of COVID-19 in schools. The authors predicted that 1925 schools would be infected in July 2020, and compared this to the PHE data (116 positive covid incidents from mid May to end of June). The authors suggest that the discrepancy in their results and the observation, is due to asymptomatic cases, poor reporting and reporting errors (line 634), while all these limitations in the PHE data could be present, the authors have not reflected on any limitations of their method.

Instead, the authors compare their results on the mean number of infected schools in England in September to the DfE report on the number of students absent from school due to self-isolating from a covid-19 contact within school (line 720). The authors find an agreement between their model output (3,087 infected schools) and the DfE data (3,000 children isolating due to an infected contact), however this is fundamentally incorrect. We are under the impression that their model doesn't project results of self-isolation within schools, and instead their model looks at the number of infected persons within a school. It is not clear what the link on line 709 is which allows the authors to equate prevalence to self isolation. This is comparing two different data types, and indeed it should be expected that the number of children self isolating is far higher than the number of infected individuals

within a school. Especially since the control measures implemented in the UK resulted in entire year groups being sent home to self-isolated when there was a single positive case identified. Additionally, the authors do not also consider the DfE data on the number of staff isolating due to a contact with an infectious covid-19 case in school, which is also available. It is unclear why the authors didn't use data from DfE (same source) on the number of pupils, teachers and other staff who are absent with a confirmed or suspected covid-19 infection. This data could be used to approximate the number of schools with an infected person in the UK, and would be a better comparison to their model output.

Response (24): The comment concerning line 709 is resolved by modifying the second of the pair of linked sentences as follows:

ONS data indicates the prevalence in Primary School age children in September – October 2020 was about 0.16%, i.e. roughly midway between our mid- and upper prevalence scenario assumptions. Given our model indicated single cases in the great majority of those schools with any infected person, jointly our pair of original scenario projections encompassed the eventual reported count of roughly 3,000 Primary Schools with self-isolating pupils in late September 2020.

Our difficulty in acceding to the several suggested changes outlined in this major comment (e.g. using DfE data) rests in the fact that, at the time we conducted our study (i.e. May - July 2020), most, if not all, the sources of observations and data the reviewer alludes to were not available. In preparing an account of our study, we were seeking to promptly report our analysis and findings, as these stood at the time the work was done (i.e. June – July 2020). It is unfortunate that the manuscript was not reviewed until several months later when, of course, much more information had emerged.

At a late stage, after our manuscript had been finalised, we did look carefully at the publicly available data from the DfE for September 2020 and (as noted by the reviewer) concluded that there was insufficient information to confidently infer the number of children absent from school with Covid-19 infection. Thus, we felt it appropriate to make simple estimates of the order of magnitude of such numbers. We think the reviewer may have mis-interpreted these. We estimated that “roughly 3,000 Primary Schools could have had self-isolating pupils in the latter half of September 2020” (ln 736), but we were unable to confidently estimate the corresponding number of pupils with Covid-19 infection. It was not reported how many classes in these schools had self-isolating pupils. If we assume one child with symptoms leads to a class self-isolating and that each school has one self-isolating class, then we obtain a lower bound of 3000 infected pupils. This said, we recognize that: (a) some self-isolating sets will involve more than a single infected person; (b) asymptomatic infection is likely to go undetected, and (c) that there may have been multiple schools with more than one self-isolating individual. We did not estimate numbers self-isolating from the data (nor did we predict these values), but note that a ‘typical’ class size of 27 pupils would give a value of ~81,000 in self-isolation from 3,000 infected schools -- a value that is an order-of-magnitude greater than our estimate. Without comprehensive reports on school attendance, it was difficult to make precise quantitative comparisons; however, we feel our real-time order-of-magnitude comparison is justified and provides some confidence that our model computed useful projections, absent any other basis for enumerating expectation values.

It is our view that the paper communicates the contribution that stochastic uncertainty science can make to policy considerations at a time of urgency when little or no coherent data exists.

Ambiguity in the methods

It is unclear if the analysis by the authors is computed over the whole month of June, July and September; over the whole term or for a single date. Line 655 (page 20) suggests it is the latter and only three single dates of the ONS prevalence sample survey are considered, namely Monday 1st June 2020, Friday 5th June 2020 and Thursday 10th September 2020. This is a major issue, as these dates aren't representative of the whole time frame, especially given the authors conclude that prevalence in the whole of September was 2.5 times higher than the prevalence in June. This is repeated throughout the text, when referring to “june-levels”, “june prevalence”, or “september-levels” in Section 6, Results; as such it is unclear if Table 2a shows the number of infected schools at the start of the month, totalled over the whole month or the whole term. This needs to be made clear in the text and in the caption of all tables and figures.

Response (25): We have modified the wording of the Abstract and the Introduction to make it clearer that our model was used to compute “snapshot” projected school infection estimates as at two nominal school return dates, viz, 1st June 2020 and 1st September 2020. Table 2a is also modified to clarify these dates.

Further detail is required to explain why data from June 5th was representative, line 322 suggests that it is the maximal value rather than a representative one. While not perfect, have the authors considered taking the average prevalence in June which would be more representative than considering a single day? Indeed, reporting errors are not consistent throughout the week in the UK, with fewer infections being recorded over the weekend (e.g. Friday-Monday), and a much higher reporting rate on Tuesdays. Taking the average in June, could minimise these errors. Additionally, the authors need to support their decision of selecting September 10th; in England, all students were welcomed to return to schools from September, but many schools chose to have a gradual return, particularly for nursery and Year 1. This gradual return is not unique to 2020 and occurs every year in September over the first 2-3 weeks (i.e. including 10th September). For scenario III, if the authors did not consider the delayed reopening, they are likely overestimating the number of infected persons in schools.

Response (26): Given limited access to ONS datasets at the time, we chose to use their 5th June 2020 prevalence as a reference value for our model calculations. As we discussed in the same para, the prevalence only fluctuated modestly in the period June – August 2020 and we felt the 5th June figure was suitable representative. In principle, while fluctuating prevalence could be incorporated in a time-stepping model, our aim was to produce tractable snapshot marker estimates for certain selected dates (which we felt policy makers would find more accessible than comprehensive time-varying modelling, with further compounded and complex uncertainty estimates).

As discussed in Section 7, prevalence data for 10th September 2020 became available from ONS and thus represented a suitable reference benchmark for the school restart period, just in time for us to run a quick counterpart check on our model projections. Again, we modify the wording of our text to reiterate that we were estimating infection levels for a single nominal day which we held to be representative of an extended school return period.

On a related note, on line 321 the authors refer to the temporal trends being noisy in the ONS prevalence data, as so do not present a temporal analysis. However, if your model can be evaluated daily to provide temporal trends, such as the number of infected schools over time, even if it is noisy, we think this would be very beneficial to the reader. This could be presented as a time series figure, demonstrating how infection in schools projected by the

BBN model changes over time, depending on the prevalence inputted. Additionally, this calculation can then replace the questionable approximation on line 622, which finds the total number of infected schools in June and July by extrapolating the results evaluated on 1st June, using the daily rate of new infected schools (rate =55, line 629, not referenced in text!) multiplied by the average infectious period. This approximation not only uses a rate which is not explained in the text, but also it does not take into account that infected individuals within a school would be isolated, and so the infectious duration of a school is not the same as the infectious duration of an individual. Instead, the authors could run their model for each day in June and take the summation, as well as presenting a time series visualisation of this data.

Response (27): With respect to taking into account mitigation measures for infected persons in a temporal analysis -- which the reviewer says would be beneficial to the reader -- some of the present authors are currently finalizing a time-stepping agent-based infection transmission model for schools, which we hope to release very shortly. Accordingly, we have added the following text at the end of Sect 6.3 where infection issues are discussed:

The rapid single-day infection snapshot model, described in this paper, can be extended to take into account mitigation measures for infected persons in schools. In addition, it can be implemented as a time-stepping agent-based infection transmission model, accounting for daily temporal variations of numbers presenting in schools as asymptomatic or infected persons; this is on-going work, to be reported elsewhere.

Thus, while we wholeheartedly agree with the reviewer that such model enhancements are very desirable, and are working on them, we feel that it is not viable to develop this suggestion within the present contribution, which records what was done at an early stage of the crisis.

At line 629, the daily rate of new infected schools (i.e. 55) is taken from Table 6; we add a cross-reference in text.

On line 252, the authors describe using incidence as a proxy for prevalence. This is not a recommended approximation, please reference in the text papers which present the limitations of this method. In particular, the authors state "to first order, prevalence can be related to incidence since, in a steady state, numbers of recovered individuals who are no longer infectious will balance numbers of new infections" on line 256, and we are under the impression that this approximation is only true where the disease has life long immunity. Life long immunity is not thought to be true for COVID-19 and the length of the waning immunity is disputed. Additionally, the author's statement is only reasonable at steady state, but we don't think it is reasonable to assume that the UK was at a steady state in June 2020. If the authors are going to take this approximation, we would like to see some sensitivity analysis performed to determine the impact of this on your results. Additionally, even assuming this proxy, it is still unclear how incidence was used to calculate prevalence. Details of this transformation should be commented on in the main text (1 sentence max), in particular was the average infectious period used to evaluate prevalence from incidence? Was the delay in reporting new cases, or the delay in presentation of symptoms considered in this approximation?

Response (28): The steady-state and incidence-prevalence proxy issue was also raised by Reviewer #1 at their Point (6), above. Noting here that, enforced by an absence of available

or accessible prevalence data, our approach to incidence-prevalence conversion is described in Appendix A1, we responded to Reviewer #1 as follows:

This [steady-state assumption] was another assumption that we felt was justified in the circumstances as it allowed our calculations to be tractable on an urgent timescale.

Text now reads:

To first order, prevalence can be related to incidence since, in a steady state, numbers of recovered individuals who are no longer infectious will balance numbers of new infections (while this is a questionable assumption, it allowed immediate model calculations to be tractable). Thus, local or regional prevalence can be scaled from local or regional incidence data: the basis for this conversion is outlined in Supplementary Information Appendix A1.

In Appendix A1: Incidence and Prevalence, the authors use a factor of 6x to reconcile incidence and prevalence, however this value has not been referenced. While the authors address that there are uncertainties in method, they have not illustrated the potential impact of these uncertainties. We think you need to perform a sensitivity analysis for this factor.

Response (29): We concur with the reviewer that this a fraught topic in many regards. As mentioned in Appendix A1 “Incidence and prevalence”, the reconciliation factor of 6x is inferred empirically in order to adjust for under-reporting of incidence (as recorded in Table A1.3). We go on to say, in Appendix A1, that “an informative proxy model for regional prevalence variation does not require an accurate knowledge of how prevalence and incidence are related because incidence varied by about three orders of magnitude at the UTLA scale.”.

We believe this adequately deals with the comment about the 6x factor.

With regard to the associated uncertainties, these are reported as sample error estimates in Table A1.3 and incorporated in the related BBN node distributions. While we agree a sensitivity test would be desirable, Reviewer #2 goes to say in the next comment that we present “a large collection of results” and criticizes the length of the manuscript; therefore, we felt we had to be selective about which results to report and discuss, and this sensitivity test was one of many which we passed over for reasons of length (and the reader’s patience).

Condensing the script, removing exploratory analysis, readability

As currently written, the manuscript is very repetitive. This is one of the main factors contributing to its length. We recommend that the manuscript be edited down to a more reasonable length, which should be achievable if the sections repeating what has already been said are removed. For example, using incidence data on UTLA as a proxy for prevalence is explored several times in the text. Due to its length, it is very difficult to find the main findings. Instead, the manuscript reads as a large collection of results, without much thought being given to which ones are most important.

Response (30): We are puzzled by this criticism. UTLA data as a proxy for regional prevalence variations is dealt with mainly in Appendix A1, being mentioned once in main text Section 4 (Data sources), once in Section 6.4 (Spatial variations in prevalence), then in Section 8 (Further work) and lastly in Section 9 (Informing policy decisions). We not accept the issue is explored several times in text, and is invoked only where needed for the narrative.

In presenting our findings as risk analysts, we endeavoured to be neutral as to our model findings and prefer to leave it to others to judge their epidemiological importance.

As mentioned above, a lot of the initial analysis/ exploration is just a subset/ specific version of later results. These earlier results, while useful in your exploration, are not required in the manuscript, and just add to the already long manuscript. Large sections of the manuscript could also be moved to the appendix, for example, section 6.3 and table 3. Similarly, why does the paper initially focus on using the mean school size, when you point out that schools vary significantly (page 18, line 581). This work could be significantly condensed by removing all exploratory analysis using the simplified assumption on the mean school size, and only presenting the analysis with school size drawn from the national distribution of school sizes.

Response (31): We regret to say we do not accept these suggested changes are justified. Table 3 comprises just three lines of very informative data from an what-if? scenario (with other related sensitivity results omitted to save space). In our rejoinder to Reviewer #1 Point (6) above, we explain our reasons for presenting these few tests of alternative scenarios on the grounds that they illustrate the sensitivity of our findings to reasonable alternative parameterisations, given the many intrinsic uncertainties in the problem.

The reviewer asks: “*why does the paper initially focus on using the mean school size, when you point out that schools vary significantly?*”. We have learned, in other societal hazard and risk domains, that when communicating complex linked uncertainties to non-specialist audiences (e.g., policy makers), gradually building-up the complexity of the model, the details of analysis and the scope of results, is advantageous. Therefore, we started by making initial calculations and analysis using a fixed school size and, for this, the mean is a suitable marker value. This allowed us to explore the influence of epidemiological uncertainties separately from the additional effects of substantial differences in the school populations. Once these effects were elucidated, it was then possible to infer the consequences of differing class sizes, and this is explored in §6.3, where we contrast the probability of infected persons in small rural schools with large urban schools, with other factors fixed. This amalgamative approach is extended in §6.5 where we explicitly use the national school size distribution in our model and further consider differences in prevalence and incidence between schools in rural and urban settings.

Additionally, we are concerned about the number of times the appendix is cross referenced. Many of the important assumptions, parameters and methodology of your work are in the appendix, making it a difficult paper to read. In particular, parameters (e.g. infection periods) should be listed in a table in the main text, as well as information of schools such as on page 7, line 280 average pupil headcount. If you have to repeatedly cite the supplementary material, then it should be contained in the main text.

The following sections should be reformatted (condensed) and moved to the main part of the paper:

- Include a table of data sources
- Table of epidemiological parameters and sources
- Couple of sentences of how prevalence data was obtained from incidence data (what parameters you used)
- Move assumptions into the main text

Response (32): we fail to see how these several suggestions are reconcilable with the preceding suggestion that there is a need to move large chunks of main text to Supplementary Information appendices because the text is too long. We believe we have partitioned our material rationally, with results and findings articulated in the main text where the reader would expect to find them, and other, less central supporting information and explanations

provided in Supplementary Information. We do not propose to undertake this major, wholesale restructuring and extensive re-writing.

The following sections should be removed from the main text and placed in the appendix:

- Results: all results using a fixed school size for all schools. Re-create Figure 2 for school sizes drawn from a distribution.

- What if? Sensitivity tests: move this section and all tables to appendix. Replace with a single sentence referencing this work with its main finding (lines 414-416) that adult prevalence has a stronger effect on projected infection numbers.

Rather than considering the comparisons to true recorded data at the end, include the prevalence in September being 2.5 times greater than the June prevalence from the beginning and replace the “scenario 4x” to “scenario 2.5x”. There’s no need to consider the “4x” or “1/4x” scenarios when we know the ratio was actually 2.5.

Response (33). We did not know this at the time our study was in progress and explain the rationale for 4x and 1/4x scenarios in our response to Reviewer #1’s Point (5), above.

It is our contention that reporting “what-if?” scenarios are not only valuable but are central to the use of uncertainty models in emergencies. Here we are reporting the way our study was performed in circumstances of high uncertainty, forecasting the likely level of infections in schools prior to empirical data becoming available. We have no doubt that tuning the model using observations would provide *post hoc* results that better match the situation that occurred, but this later tweaking gives a false sense of the utility of the model. We believe we have demonstrated that simple “what-if?” scenarios can be rapidly constructed that provide informative projection, even when data is scarce.

In your spatial analysis, does each half-log prevalence bin correspond to a specific region in the UK? This could be important in terms of the number of students eligible to attend school in June 2020 varying spatially. We also have the following concerns with your spatial analysis:

- Page 15, line 497, not clear what six half-log incidence bins with sampling uncertainties are. My understanding is that you are dividing England into 6 groups based upon the prevalence in UTLA, and table 5a shows the number of UTLAs in England which has a very-high prevalence level in bin 6 and UTLAs with low prevalence in bin 1? We would name each bin by categories which are more intuitive such as “very-high prevalence”, “high”, “medium”, “medium-low”, “low” and “very-low”.

Response (34): The reviewer’s understanding of the basis is correct: we divided England into six groups based on UTLA prevalence levels. However, we do not subscribe to using ambiguous and ill-defined verbal category appellations when these can be expressed as definite numerical ranges (we know, from experience with volcanoes and climate change, that disastrous consequences can ensue from such vernacular descriptors).

- This isn’t really a spatial analysis. You are considering 6 independent regions with different community prevalences, rather than a number of potentially interacting populations (e.g. a low prevalence region surrounded by high prevalence regions could behave differently to a low-prevalence region surrounded by low-incidence regions). As far as we can tell, you have essentially just performed a sensitivity analysis on the community prevalence.

Response (35): In essence, the reviewer is correct, this is not a spatial analysis *sensu stricto*. However, our purpose is to indicate first order geographical variations of schools’ infection

rates within England and believe the reader will understand the rubric “spatial” in this context.

- We don’t think Table 5a is needed in the text (move to appendix) instead just include in the main text the population weighting of each category.

Response (36): Reviewer #1 at Point (18) above recommended we modify this table to prevalence rates per 100,000 persons; we infer Reviewer #1 felt it is useful.

- For the spatial study, Page 15, line 500 - have you calculated the number of positive tests or the proportion of positive tests (i.e. positive/(positive+negative)). Given the heterogeneous nature of testing in England, the analysis needs to be done on the proportion of positive tests.

Response (37): It is the latter.

Referencing literature and sources

The manuscript currently looks very little into the existing literature on covid-19 in schools. The work would greatly benefit from a review of the current literature. As such, a section describing how this work compares to more traditional epidemiological models, and how the authors’ work fits into the literature would improve the manuscript. As it stands, almost a quarter of the references are from the authors’ works - is this representative of the covid-19 literature?

A large number of parameters are introduced, without any citations for them. If the values are not taken from a particular source, a justification (and ideally a sensitivity analysis) should be provided for them. For example, line 203, you include a +/- 10% uncertainty on the estimates. Why 10%? Is this a standard value to take? It’s not clear that this will be sufficient to cover the part time staff and teaching staff on rota. There are many other similar occurrences in your manuscript. As stated elsewhere in this review, this would be most convenient listed in a table along with all the other parameters you’ve used, rather than scattered across the main text and appendix.

There are a number of other places where citations should be included for the claims you have made, including but not limited to the following:

- Page 3 of 27, line 77, cite to references
- Page 4 of 27, line 103, cite other studies on this topic
- Page 6 of 27, line 193, Primary and secondary, and establishment type (state, private etc) is available on the Gov website.
- Page 6 of 27, line 211, reference where you got the susceptibility ratio
- Page 6 of 27, line 184, how do you calculate your contact matrix. Provide reference to other contact matrix surveys. What is your contact matrix?
- Page 7 of 27, line 240, reference the typical absenteeism value
- Page 9, line 318, the value of 65% staff returning to school in June, cite this. Or is this an approximation? If latter, what was actually observed?
- Page 15, line 492, citation needed
- Page 15, line 495, citation for the regional prevalence in high-risk regions being 350 times worse

Response (38) These comments suggest to us that Reviewer #2 has in mind we should develop a very different piece of *post hoc* (research) work, quite different from our account which sought to describe a rapid and pragmatic quantitative assessment of primary schools’ infection risks for immediate decision support at the earliest stages of a novel pandemic outbreak (when many of the data sources and references mentioned did not yet exist in the public domain). Therefore, we cannot agree to make these additions and changes.

Figures

● Figure 1: Why is children_returning_count in green? What is the reasoning behind the stdNorm_distribution? The quality of this figure is very blurry.

Response (39): The green node outline has no meaning and will be modified in the final version; the plot quality is determined by the originating software package and will be improved, if feasible.

● Results: figure 2, remove the border of this figure and include the mean/std as a legend in the plot. Provide y labels and x labels. Improve image quality (very blurry). This figure shows the distribution for Scenario IIIb, we think your results section would improve greatly if you included this figure for each of the scenarios.

Response (40): Again, plot layout and quality is dictated by the generating software package and is not amenable to editing or enhancement. Adding nineteen additional counterpart plots for all the scenarios that comprise the results will hugely lengthen the manuscript; all the statistical attributes of these scenario distributions are reported in Tables 2a and 2b.

Other

We have included below a number of other suggestions that we didn't think neatly fell into one of the above categories.

Could you clarify how you have produced table 2b in your results section? Is this the number of primary school children you would expect nationally or per school? We assume the former, if the former (over the whole of England) changes this table from the number of pupils to the proportion of pupils, teachers and ancillary staff. It would be interesting to know the proportion of teachers rather than raw numbers as this can be compared to the national prevalence. Again, change tables 3 and 4 to use proportions.

Response (41): Yes, Tables 2b, 3 & 4 relate to counts nationally; the captions have been modified accordingly. We do not see a need to change Tables 3 and 4 to proportions.

Could you clarify what table 5b shows? Is the spatial prevalence in this table showing the number of primary schools in England after using a different prevalence for each binned region? Table 6 is far more interesting, presenting the proportion of schools in each binned category which had 1 or more infected case, not just the total in England. Table 5c is comparing a non-spatial prevalence to a spatial-prevalence method, without providing any insight on the spatial heterogeneity of covid cases within schools. As such, this should be moved to the appendix.

Response (42): Table 5b illustrates the effects of introducing first spatially varying prevalence into the calculational model (row 2 "Spatial prevalence") and then adding school size distribution to the spatial prevalence model (row 3). We think this allows the reader to judge the relative effects of extending the modelling basis. Table 5c provides a similar exposition and we do not agree it should be removed to the appendix.

We are also concerned that in Table 6, you do not consider the total number of pupils, teachers, staff in each of the binned sections and assume that this is constant for all binned sections? We would like to see this carried out for a non-average sized school. Likewise with Table 5c, rather than aggregating your spatial results over the whole of England, it would be more interesting to present the number of infected persons for each binned region. Change table 5c to proportion of pupils in each binned region which are infected, and move to the appendix.

Response (43): The caption to Table 6 states: “calculations are based on the assumption that the total numbers of schools per prevalence bin are proportional to the corresponding population size” and we think it reasonable that the reader will understand that numbers of pupils, teachers and staff are proportional pro rata as the population sizes. Extending the analysis to a non-averaged size school (what size?) would entail making the text and discussion yet longer.

On page 18, line 585, you say there is no change in the overall number of infected persons when taking the mean school sizes vs school sizes drawn from the national distribution however there is no table to back up this statement. We think it would be best to remove all work using the mean school size, and just focus on national distribution.

Response (44): We think line 585 states the most likely, and probably only, explanation for obtaining the same results from the two ways of characterizing schools sizes. We disagree about removing mean school size calculations from the manuscript – we simply report what was done at the time in an on-going data acquisition and analysis process that had to be iteratively developed and refined as fast as circumstances allowed. Furthermore, as noted earlier, from the perspective of uncertainty communication we believe there are benefits in building-up the complexity of model and the interacting sources of uncertainty, rather than opting only for the results from the most complex model setup. Attempting to convey such all-embracing findings to non-specialists, perhaps very unfamiliar with probabilistic assessments of complex issues, is – in our experience -- a fraught policy.

On page 19, could you clarify how the mean probabilities were found? Remove all lists of $N=0$, Prob = ... into a single table, so there is a column in the table for each scenario considered.

Response (45): text changed from “found” to “our model computed”.

Minor

- There is no Table 1? Document starts with Table 2a, although the generic scaling factor is in table 1 (page 15, line 499)

Response (46): Corrected as reference is to Appendix A1 Table A1.1; other main text tables are now renumbered.

- Page 3 of 27, line 48 and 49 “The devolved administrations... re-open their schools.” Don’t think this is necessary

Response (47): Removed

- Page 5 of 27, line 140, assume the audience does not know what an “acyclic graph” is.

Response (48): Explanation added.

- Page 5 of 27, line 143 why is the BBN method only “quite general”

Response (49): The approach is general in application ... “quite” removed.

- Page 6, line 213 this reference to background about BBN should be in section 3. Methodology where BNN is introduced

Response (50): Text moved, as suggested.

- Page 9, line 306, add to this paragraph what the figure shows, i.e. the distribution of the number of primary schools with at least one infection.

Response (51): Para modified as follows:

An example model output distribution is shown in Figure 2, showing the distribution of number of schools with at least one infection under Scenario IIIb (above). This plot illustrates one BBN result with typical uncertainty spread and some skewness in distribution shape; such long-tailed behavior, as evinced here, could have implications for policy decision-making. This is pertinent because a societal risk assessment that is expressed solely as a

central tendency estimate (e.g. as a mean or median, alone) may not be adequate or appropriate for high-consequence risk-informed decision making. Therefore, it is crucial to express such assessment results with a suitable set of confidence levels or exceedance probabilities, such that the decision maker can have a full probabilistic context for appraising the conservatism and tolerability, or otherwise, of the threat.

- Page 12 line 363, sensitivity analysis is common amongst many epidemiological models and isn't unique to BBN. Rephrase/remove lines 363 to 365.

Response (52): While this statement is true in principle, the key point here is that UNINET (not BBNs in general) allows direct conditionalisation testing via its graphic interface, with no need for additional coding or subroutines. Text is modified to read:

One of the powerful analytic features of the UNINET algorithm underpinning our model is 'what-if?' tests can be implemented directly via its graphic interface, enabling the re-computing model results based on specific parameter values or ranges of values without recourse to additional coding or subroutines. The program can accommodate a single conditionalization that is applied to one variable, or several different conditionalization restraints applied jointly, and simultaneously, to a set of variables in the model. For any parameter, the analyst defines which value or range of values to use for the what-if test of interest.

- Move the results on the probability of having two or more cases, or three or more cases to earlier. The rest of section 6.3 can be moved to the appendix.

Response (53): We have looked at what is involved in doing this and regret we feel unable to accede to the suggestion – we judge the amount of work and risk of engendering editorial and narrative infelicities are not justified.

- As a related aside, in figure 1 a fixed input value for prevalence is taken, I wonder whether the confidence interval of prevalence should be considered here.

Response (54): This not correct: node *<Adult_prevalence>* is a random variable node which carries an uncertainty distribution through into the calculations (see caption).

Appendix E

Manuscript ID RSOS-202218: Pupils returning to Primary Schools in England during 2020: rapid estimations of punctual COVID-19 infection rates

Authors' response to RSOS Editor

By email dated 24 August 2021, the RSOS Associate Editor (Prof. Ross) wrote:
I would like you to consider adding a sensitivity analysis to the paper, in which you assess both the incidence-prevalence and steady-state assumptions. It is my understanding that knowing the impact of these assumptions would be useful to guide the potential use of your framework in future outbreaks; and that your decision to not do this was motivated by Reviewer 2 also wanting you to shorten the length of the paper.

We have responded first by re-captioning Section 7 and entering a new first sub-section header, as follows:

7. Checking the model

7.1 Reviewing scenario projections

..... this is followed by the existing text, as before.

Following sub-section 7.1, two new subsections are added in respect of the editor's suggestion:

7.2 Incidence-to-prevalence conversion factor - sensitivity analysis

As discussed in Section 4, we examined regional incidence data – available from June 2020 – and used these as ergodic analogues for prevalence. On this basis, we derived a simple relation between incidence and prevalence, leading to the assumption that the latter can be considered approximately proportional to 6x incidence under steady-state infection-response conditions at the time (per Supplementary Information Appendix A1). In our stochastic model, this incidence multiplier is represented by an empirical two-parameter Lognormal uncertainty distribution with scale = 6 and shape determined by combining sets of geographic-referenced incidence data (Supplementary Information Appendix A1).

In Table 7a, we examine the effect on our estimates of national numbers of infected pupils and infected teachers under Scenario IIIa (see also Section 6.1 and Table 1b) by re-running our model with Lognormal prevalence scale changed to 4 or 8, instead of 6, while retaining the same shape factor.

Table 7a. Sensitivity analysis for alternative incidence-to-prevalence conversion factor – effect on estimated numbers of infected pupils and infected teachers nationally in Primary Schools in England, for return to school in September 2020 -- Scenario IIIA (see also Sect. 6.1 & Table 1b). n.b. while modelling results are reported to nearest whole number, such precision is not claimed for these indicative projections.

Model	Mean number	5% quantile	50% quantile	95% quantile
Scenario IIIa basis model: Sept 2020 return & inferred June 2020 prevalence (per Table 1b); incidence-to-prevalence factor 6x				
Children	1417	468	1203	3090
Teachers & TAs	249	102	223	487
Scenario IIIa with incidence-to-prevalence factor 4x				
Children	944	356	828	1927
Teachers & TAs	166	79	153	295
Scenario IIIa with incidence-to-prevalence factor 8x				
Children	1888	711	1655	3855
Teachers & TAs	331	159	306	589

Overall, the means of these sensitivity results directly reflect the corresponding changes in prevalence scaling, as might be expected in an essentially linear model. The quantiles of the associated stochastic distributions also reflect the scaling factors, albeit with minor variations likely due to stochastic sampling effects in the different computational runs with UNINET.

7.3 Incidence / prevalence ‘steady state’ assumption - sensitivity analysis

In Section 4, the basic assumption underlying the incidence-to-prevalence factor in our model, just discussed, was that COVID-19 incidence and prevalence could be presumed to equilibrate into steady-state statistical processes at national and regional scales and that this would remain stable from June 2020 (i.e. the incidence data then available) through to return-to-school in September 2020. The possibility existed, however, that infection processes or incidence rate itself could evince trend changes with time, such that our assumption of a constant incidence-to-prevalence factor would be questionable. Therefore, it is reasonable to ask about potential effects on our results if the steady-state assumption were not fully valid.

Bearing in mind our model simply presents a nominal single-day future projection of in-school infection numbers, here we take advantage of UNINET’s conditionalized re-sampling capabilities to test the impact on our projections of upward or downward systematic changes in prevalence over the forward period from June to September 2020, again using Scenario IIIa results from Table 1b as reference. To demonstrate the sorts of sensitivity analysis that can be undertaken, we re-run the projection model with two conditionalizations applied: (a) prevalence from June 2020 might have followed an upward trajectory, falling somewhere within its 50th to 95th percentile uncertainty range; and (b) a downward trend in prevalence might have occurred, falling within its 5th to 95th percentile uncertainty range. Full stochastic re-sampling leads to the results summarised in Table 7b.

Table 7b. Sensitivity analysis of Table 1b Scenario IIIa projections for in-school infection numbers in September 2020 if, in the interim, prevalence departed from the June 2020 basis model steady-state assumption. n.b. computed results are shown to nearest whole number, but such precision is not claimed for these indicative projections.

Model	Mean number	5% quantile	50% quantile	95% quantile
Scenario IIIa basis model: Sept 2020 return & inferred June 2020 prevalence (per Table 1b); incidence-to-prevalence factor 6x				
Children	1417	468	1203	3090
Teachers & TAs	249	102	223	487
Scenario IIIa with prevalence increasing within 50th to 95th percentile uncertainty range				
Children	1730	890	1609	2990
Teachers & TAs	304	236	292	411
Scenario IIIa with prevalence decreasing with 5th to 50th percentile range				
Children	1002	516	940	1706
Teachers & TAs	176	129	177	218

With the modelled general upward trend in prevalence, the infection number distributions for Children and Teachers & TAs are both skewed with longer upper tails (Table 7b). For the conditional numbers under a general downward trend in prevalence, the Children’s distribution is still marginally skewed with a slightly longer upper tail, whereas the Teachers & TAs distribution support is reversed, more toward smaller numbers.

These two sensitivity analyses illustrate how our model can be interrogated for the impacts of different circumstances, conditions or parameterizations. While the number of alternative scenarios that can be quantified probabilistically in this way is almost limitless, such sensitivity tests are best tailored to the policy concerns of the problem owner or decision maker.

A new paragraph is added at the end of **Section 9 Informing policy decisions**, reflecting the role of such sensitivity analyses for policy setting:

The two sensitivity analyses, described above (Sections 7.2 and 7.3), serve to illustrate how our model can be interrogated for the impacts of different circumstances, conditions or parameterizations. Such sensitivity tests can be customized exactly to the concerns of the problem owner or decision maker for policy setting in any future societal infection wave or new outbreak. In this regard, an infection transmission algorithm and model diagnostic capabilities have been extended to accommodate the dynamics of new variant coronavirus infections in schools, allowing efficacies of alternative mitigation measures to be gauged objectively.

Finally, we have slightly modified **Acknowledgements** to read:

We thank Leon Danon and Jonty Rougier for commenting on an earlier version of this paper. We are grateful to the RSOS editors and two anonymous reviewers for many constructive and insightful suggestions that helped significantly improve this contribution.